# Seismogenic potential and tsunami threat of the strike-slip Carboneras Fault in the Western Mediterranean from physics-based earthquake simulations

José A. Álvarez-Gómez[1], Paula Herrero-Barbero[1,2], and José J. Martínez-Díaz[1,3]

[1]Department of Geodynamics, Stratigraphy and Palaeontology. Faculty of Geology. Complutense University of Madrid, Madrid, Spain
[2]Geosciences Barcelona CSIC, GEO3BCN-CSIC, Barcelona, Spain
[3]IGEO, Geosciences Institute, CSIC-UCM, Madrid, Spain

**Correspondence:** José A. Álvarez-Gómez (jaag@ucm.es)

**Abstract.** Strike-slip fault ruptures have a limited capacity to generate vertical deformation, and for this reason they are usually dismissed as potential destructive tsunami sources. At the western tip of the western Mediterranean, in the Alboran Sea, tectonics is characterized by the presence of large transcurrent fault systems and minor reverse and normal faults in a zone of diffuse deformation. The strike-slip Carboneras fault is one of the largest sources in the Alboran Sea, and therefore, with the greatest seismogenic capacity. It is also one of the active structures with higher slip rates in the Eastern Betic Fault Zone and has been proposed as source of the damaging 1522 (M6.5; Int. VIII-IX) Almeria earthquake. The dimensions and location of the Carboneras fault imply a high seismic and tsunami threat. In this paper we present tsunami simulations from seismic sources generated with physics-based earthquake simulators. We have generated a 1 Myr synthetic seismic catalogue consistent on 773,893 events with magnitudes ranging between $M_W$ 3.3 and 7.6. From these events we have selected those sources producing a potential energy capable of generating a noticeable tsunami, being earthquakes with magnitudes ranging from 6.71 to 7.62. The Carboneras Fault has the capacity to generate locally damaging tsunamis, however, on a regional scale its tsunami threat is limited. The frequency – magnitude distribution of the generated seismic catalogue reflects the variability of magnitudes associated to the rupture of the entire fault, departing the upper limit from the classical Gutenberg-Richter potential relation. The inter-event time for the maximum earthquake magnitudes is usually between 2,000 and 6,000 years. The use of physics-based earthquake simulations for tsunamigenic sources allows an in-depth characterization of the scenarios, allowing a qualitative leap in their parametrization.

## 1 Introduction

Tsunamis are generated by any natural event that involves an immediate alteration of the elevation of the free surface of the sea. This alteration may be due to events that directly alter the sea surface (usually meteorological, meteoric or volcanic events) or by geological events that abruptly modify the ocean floor (earthquakes or submarine landslides). Earthquakes are the geological events that most often generate destructive tsunamis (NGDC, 2022), and this ability depends on their magnitude and depth, as well as on their mode of seismic rupture (e.g., Burbidge et al., 2015; Geist, 1998; Gibbons et al., 2022). The

rake, the orientation of the slip vector on the fault plane during seismic rupture, is one of the most determining parameters in the generation of tsunamis, presenting the thrust and normal faults, with dip-slip rupture, the greatest capacity. On the other hand, the strike-slip ruptures, with rakes close to the horizontal, have a limited capacity to generate vertical deformation on the seafloor, and for this reason they are usually dismissed as potential destructive tsunami sources.

Although the lower capacity of strike-slip faults to generate tsunamis is a proven fact, it is not negligible, as has been numerically demonstrated (Elbanna et al., 2021; Legg et al., 2003; Tanioka and Satake, 1996; Ulrich et al., 2019) and observed (Baptista and Miranda, 2009; Frucht et al., 2019; Gusman et al., 2017; Heidarzadeh et al., 2017; Ho et al., 2021), occasionally linked to submarine landslides also (Hornbach et al., 2010; Xu et al., 2022). This is of special relevance in local sources, where the dispersion of the tsunami waves is low, and the local fault complexities and rupture-slip variations are key parameters on tsunami impact (Geist, 2002).

The study of tsunami hazard, due to the scarcity of events from a statistical point of view, is frequently approached from numerical modelling. These models are usually based on the simulation of tsunamis generated by ruptures of simple, rectangular, fault planes with homogeneous slip, sometimes mixed with more complex ruptures or the combination of several rectangular ruptures (e.g. Basili et al., 2021; Power et al., 2012; Davies et al., 2018; Zamora and Babeyko, 2016). Codes based on the Okada (1985) or Mansinha and Smylie (1971) equations are used to obtain the seafloor deformation produced by the earthquake.

However, the variability in the slip distribution on the fault plane is a fundamental parameter to understand the occurrence of maximum amplitudes in destructive events (Fujii et al., 2011; Gusman et al., 2012; McCloskey et al., 2008; Satake et al., 2013; Yamazaki et al., 2011). This variability is of special relevance in local sources, which if modelled as simple ruptures, cannot capture the complexity of the earthquake rupture process. Wave propagation and flooding are highly non-linear processes, very sensitive to local variations in shallow waters. To overcome this limitation, methodologies have been proposed based on the stochastic (or random) generation of slip patterns in faults (Geist, 2002; Goda et al., 2015; Lavallée et al., 2006; Løvholt et al., 2012; Mai and Beroza, 2002), or on the use of physical dynamic rupture models for particular events (Elbanna et al., 2021; Kozdon and Dunham, 2013; Madden et al., 2021; Maeda and Furumura, 2013; Ryan et al., 2015; Wendt et al., 2009; Wilson and Ma, 2021).

Our approach is based on the use of physics-based earthquake simulators (Rundle, 1988). These simulators have been developed in recent decades in order to overcome the temporal limitation of the instrumental seismic catalogue in probabilistic seismic hazard assessment (PSHA) (Robinson et al., 2011; Shaw et al., 2018), especially in the characterization of large events. Through the development of models based on earthquake physics, synthetic catalogues of hundreds of thousands of years can be generated whose characteristics reflect those of the instrumental catalogue but incorporating the long-term evolution of the seismic cycle and the complex interactions of fault systems (Console et al., 2018, 2015; Robinson and Benites, 1995). Moreover, recent development of numerical codes based on the rate-and-state constitutive law for fault slip and frictional behaviour (Dieterich, 1992, 1995) allows not only the modelling of long-term seismic cycle deformation, but also the short-term rupture process based on a quasi-dynamic physical approximation to the rupture propagation (Richards-Dinger and Dieterich, 2012) producing earthquake ruptures similar to those in fully dynamic models (Whirley and Engelmann, 1993).

The western Mediterranean presents a complex tectonic history and context (e.g. Chertova et al., 2014; Gómez de la Peña et al., 2021; Romagny et al., 2020); and is characterized by the development of a series of arcuate Fold-and-Thrust Belts surrounding back-arc formed deep-water basins (Faccenna et al., 2004; Rosenbaum and Lister, 2004). The current rate of shortening between the Nubian and Eurasian plates in the western Mediterranean is approximately 5 mm/yr (Serpelloni et al., 2007; Vernant et al., 2010), which, being distributed in a large number of structures, makes them to have low or very low velocities. These low deformation rates imply that the seismic cycles of the main structures are very long and the instrumental seismic catalogue can hardly show the characteristics of the potential major events.

At the western tip of the western Mediterranean, in the Alboran Sea, tectonics is characterized by the presence of large transcurrent fault systems and minor reverse and normal faults in a zone of diffuse deformation (Ballesteros et al., 2008; Martínez-García et al., 2013). These structures, formed during the Miocene in a transcurrent and mainly extensional tectonic context, were latter reactivated in a transpressional post-tortonian setting (Bourgois et al., 1992; Comas et al., 1992; Do Couto et al., 2016; Herrero-Barbero et al., 2020; Martínez-García et al., 2017). Although the reverse faults associated with the Alboran ridge seem to have the greatest tsunamigenic potential (Álvarez-Gómez et al., 2011a, b; Gómez de la Peña et al., 2022), the mainly strike-slip faults (Yusuf, Al -Idrisi and Carboneras) are the ones with the greatest length and, therefore, the greatest seismogenic capacity (Somoza et al., 2021).

The Carboneras fault is a left-lateral transpressive structure oriented N50-60ºE with a length of $\sim$ 150 km, most of them offshore (Gràcia et al., 2006; Somoza et al., 2021). It is one of the active structures with higher slip rates in the Alboran Sea and in the Eastern Betic Shear Zone (Masana et al., 2018; Moreno et al., 2015), a fault system that crosses the SE of the Iberian Peninsula forming a tectonic corridor on which a significant density of population and industry is based (Figure 1). Moreover this fault has been proposed as source of the 1522 Almeria earthquake, a damaging earthquake that reached intensities of VIII-IX in the city of Almeria (Martínez Solares and Mezcua, 2002), and possibly related to a local tsunami (Reicherter and Hübscher, 2007; Reicherter and Becker-Heidmann, 2009). The location of this event is uncertain, being located onshore in the historic/instrumental seismic catalog (IGN-UPM, 2013) (Figure 1) but with an offshore epicentral location proposed by Reicherter and Hübscher (2007). It is noteworthy that much of the instrumental seismicity in the study area cannot be directly related to the most conspicuous active faults (Figure 1). Although it can be puzzling, is a common feature in slow moving faults, where seismic cycles are on the order of thousands of years. In these cases the use of physics-based earthquake simulators can be specially useful to get insight into the seismogenic behaviour of the fault system.

The dimensions and location of the Carboneras fault poses a high seismic and tsunami risk potential. According to previous studies this fault has the capacity to produce events with magnitudes up to 7.1 - 7.4 (Álvarez-Gómez et al., 2011a; García-Mayordomo et al., 2017; Gómez de la Peña et al., 2022; Gràcia et al., 2006) and even 7.6 according to Moreno (2011), with mainly horizontal left-lateral component but some reverse dip-slip motion too (Moreno et al., 2015). Although the tsunami simulations done to date (Álvarez-Gómez et al., 2011a, b; Gómez de la Peña et al., 2022) discard major damaging events, the simplicity and assumptions of such simulations must be re-evaluated. The Carboneras Fault has also been modelled in the frame of probabilistic tsunami hazard assessments but from regional simplified approaches where this source is not specifically studied (Basili et al., 2021; Sørensen et al., 2012).

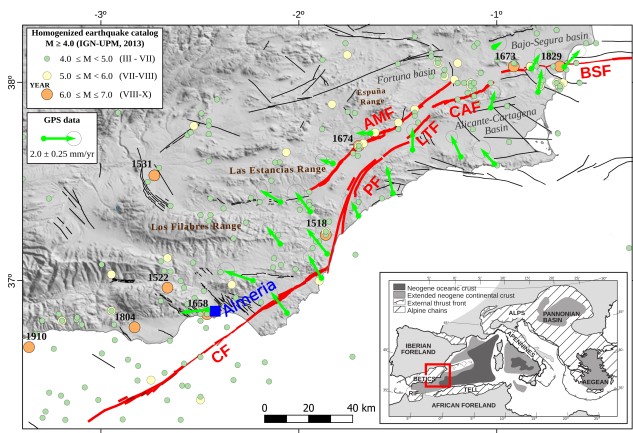

**Figure 1.** Seismotectonic setting of the Eastern Betic Shear Zone (EBSZ), modified from Herrero-Barbero et al. (2021). The filled circles show the epicentres of the declustered and homogenized earthquake catalogue from the IGN-UPM (2013) (Cabañas et al., 2015). The green arrows represent geodetic velocities from GNSS networks (Borque et al., 2019; Echeverria et al., 2013) with Eurasia-fixed reference frame. Fault traces from the Quaternary Active Fault Database of Iberia, QAFI v.3 (García-Mayordomo et al., 2017). In red the main faults of the EBSZ: Carboneras fault (CF), Alhama de Murcia fault (AMF), Palomares fault (PF), Los Tollos fault (LTF), Carrascoy fault (CAF), and Bajo-Segura fault (BSF). The inset shows the location of the EBSZ in the western tip of the Alpine orogenic belt (modified from Martínez-García (2012)). Note that the projections may differ from a Mercator projection.

In this paper we present tsunami simulations based on the generation of a synthetic catalogue of earthquakes whose characteristics resemble the instrumental and historical seismicity recorded in the area (Herrero-Barbero et al., 2021). From these simulations we make estimates of maximum wave elevations for seismogenic tsunamis and recurrence intervals for signifi-

cant events in order to reassess the threat posed by the Carboneras fault in the context of the Alboran Sea and the western Mediterranean.

## 2  Earthquake ruptures simulation

Reproducing a long-term catalogue of earthquake ruptures requires a computationally efficient approach to the physical processes that control earthquake occurrence. Earthquake simulators (Rundle, 1988; Tullis et al., 2012; Ward, 2000) are computer

codes that use fault geometry, stress interactions and frictional resistance to produce long earthquake sequences, overcoming the completeness limitations of the instrumental record. The multi-cycle earthquake simulations necessarily adopt approximations to elastodynamics to make computation feasible and, unlike fully dynamic single-event simulators (see e.g., Harris et al., 2018), seismic waves are not computed. Even so, recent modelling enhancements have successfully extended their use in more complex fault geometries (Field et al., 2014; Shaw et al., 2018) and for different representations of fault friction, rheology

and stress transfer (Pollitz, 2012; Richards-Dinger and Dieterich, 2012; Sachs et al., 2012; Schultz et al., 2018; Ward, 2012); therefore, a better validation of the quasi-dynamic part of the seismic cycle is achieved.

Our approach is based on the application of the RSQSim earthquake simulator (Dieterich and Richards-Dinger, 2010; Richards-Dinger and Dieterich, 2012). The physics-based RSQSim code reproduces earthquakes into a fully interacting 3D fault model. It performs the physical processes leading rupture nucleation and propagation through a boundary element formulation that incorporates rate- and state-dependent friction based on Dieterich (1995), in which frictional shear stress is quantified as:

$$\tau = \sigma \left[ \mu_0 + a \ln \left( \frac{V}{V_0} \right) + b \ln \left( \frac{\theta V_0}{D_C} \right) \right], \tag{1}$$

Given that this is a quasi-dynamic approximation, long-term stress accumulation and earthquake slip at each fault element is separated efficiently into three sliding states: 0) locked, 1) nucleating, and 2) sliding. The result is a long synthetic earthquake catalogue including a comprehensive and detailed record of complex earthquake ruptures with heterogeneous slip. Recent results obtained using the RSQSim code are promising in relation to potential practical applicability (Chartier et al., 2021; Herrero-Barbero et al., 2021; Howarth et al., 2021; Shaw et al., 2018, 2022).

The 3D structure of the Carboneras fault is integrated in a more complex fault model of the Eastern Betic Fault Zone (Figure 2), which includes kinematic properties of the main faults, such us slip rates and rakes (Herrero-Barbero et al., 2021). RSQSim simulations have been run on this fault system model of triangular elements with 1 km² resolution and maximum fault depths between 8 and 12 km, considering the seismogenic thickness in the area (García-Mayordomo, 2005; Fernández-Ibañez and Soto, 2008; Mancilla et al., 2013; Grevemeyer et al., 2015). The Carboneras fault is defined in the model as a sub-vertical (dipping 85º SE) sinistral strike-slip structure (Bousquet, 1979; Masana et al., 2018; Moreno et al., 2015; Rutter et al., 2012), N45º-60º strike, and segmented into two fault sections: an onshore northern section, partially offshore at the SW and connecting with the Palomares fault at the NE; and a totally offshore southern section.

The slip rate of the Carboneras Fault has been estimated by means of geodetic data (GNSS or InSAR) and geologic data (paleoseismology, tectonic geomorphology). From the geological point of view (Moreno et al., 2015) obtained a minimum strike-slip rate of 1.31 mm/yr for a section in the onshore segment of the Carboneras fault, and a minimum dip-slip rate of 0.18 mm/yr. The offshore segment of the Carboneras fault has been extensively studied also by Moreno (2011); obtaining a minimum strike-slip rate of 1.3 mm/yr and dip-slip rates between 0.1 and 0.3 mm/yr. The geodetic data obtained to date is from the onshore segment by means of high resolution GNSS campaign measurement and permanent stations, with the preferred value of strike-slip rate of 1.3±0.2 mm/yr (Echeverria et al., 2015). Taking into account that the onshore and offshore geologically obtained slip rates are the same, and that the onshore geodetic data coincides with the geological data, we have used the estimated slip-rate of 1.3 mm/yr in our models. Kinematic numerical models developed in the area obtain values in the same range for the Carboneras fault slip-rate in their preferred results (Jiménez-Munt and Negredo, 2003; Cunha et al., 2012; Neres et al., 2016).

Besides the input kinematic data (detailed in Herrero-Barbero et al. (2021)), the simulations are governed by rate- and state-dependent friction parameters, a and b, that reproduce the effect of the velocity-change on the coefficient of friction (Dieterich, 1979; Ruina, 1983). These frictional parameters have a notable impact on the slip distribution and spatio-temporal clustering

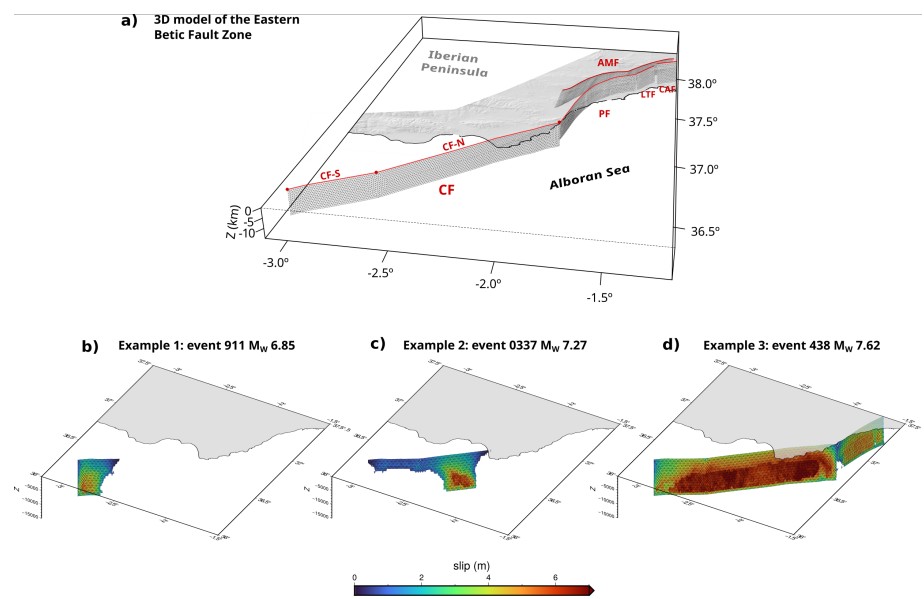

**Figure 2.** a) 3D fault model used for the synthetic seismic catalogue simulation. CF: Carboneras Fault, CF-S: Carboneras Fault southern section, CF-N: Carboneras Fault northern section, PF: Palomares Fault, LTF: Los Tollos Fault, CAF: Carrascoy Fault, AMF: Alhama de Murcia Fault; b) example of a $M_W$ 6.85 event; c) example of a $M_W$ 7.27 event; d) example of a $M_W$ 7.62 event.

(Noda and Lapusta, 2013; Richards-Dinger and Dieterich, 2012; Scholz, 1998). We define reference rate-and-state values
based on experimental data taken from a nearby location in the fault zone (Niemeijer and Vissers, 2014; Rodriguez Escudero, 2017). These experimental values are the starting point before testing several synthetic catalogs with variation of the rate-and state-dependent coefficients, until achieving the best-fit values (Herrero-Barbero et al., 2021). The aims of the testing process were to match frequency distributions with a Gutenberg-Richter b-value close to 1.0±0.1, and to correlate the synthetic seismicity with instrumental and paleoseismic data (Herrero-Barbero et al., 2021). This b-value has been estimated in the same
seismogenic zone in previous works based on instrumental seismicity (IGN-UPM, 2013; García-Mayordomo, 2015), and is also a reference value as assumption in numerous papers of synthetic seismicity modeling (e.g. Console et al., 2017; Shaw et al., 2018, 2022). Finally, a preferred set of input model parameters is selected for the best-fit catalogue: rate-and-state friction parameters a=0.001 and b=0.010 (not to be confused with the a-value and b-value of the Gutenberg-Richter frequency magnitude distribution); a steady-state friction coefficient $\mu_0$=0.6; a depth-variable normal stress with a 20 MPa/km gradient,
and a b-value of 1.05 (Figure 3). Defined frictional parameters in this study entail a totally seismogenic behaviour of this fault system, although Faulkner et al. (2003) also suggested possible creeping sections in the Carboneras fault zone due to the mechanical heterogeneity of its fault gouge. As the seismic productivity (the a-value in frequency magnitude distribution, FMD) depends on the slip-rate and seismic coupling of active faults, our approximation, which considers fully coupled structures can be considered as the maximum seismic productivity expected.

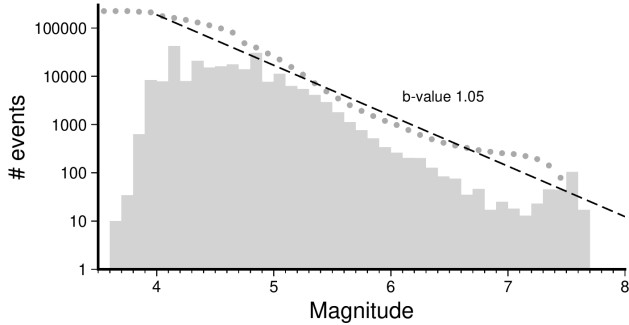

**Figure 3.** Frequency magnitude distribution of the generated synthetic earthquake catalogue for the Carboneras Fault. The grey bars show the discrete count of events, while the grey dots show the cumulative form of the distribution. The dashed line shows the best fit of the Gutenberg-Richter law.

According to the selected input model parameters, a 1 Myr-synthetic earthquake catalogue has been generated (Figure 3), from which the first 2000 years have been discarded to avoid artefacts until the simulation stabilizes. In total, 773,893 events have been obtained, with a magnitude range of $3.3 \leq M_W \leq 7.6$. The Carboneras Fault is the seismogenic source that generates the most frequent synthetic seismicity, with almost a 30% of the events in the catalogue, of which 0.6% of total events are $M_W$ $\geq 6$ earthquakes. For $6.5 \leq M_W < 7.0$ events, the most frequent inter-event time intervals range between 800 and 6,000 years

(Figure 4). Logically, the earthquake frequency decreases as the magnitude increases (Figure 4). However, from magnitude $M_W$ 7.0-7.1, the simulation shows an increase in the frequency of events, therefore the recurrence intervals of the most damaging $M_W \geq 7.0$ earthquakes would be shortened. As the inter-event time intervals depends on the seismic productivity (a-value of the FMD) our results are the most conservative (shortest inter-event times) as we are assuming fully coupled faults. The largest number of these major simulated ruptures in the Carboneras Fault is nucleated in the northern section, being physically capable

to propagate a complete fault-length rupture. Between them, 115 ruptures are also transferred to a portion of the southern branch of the Palomares Fault (Figure 2d), increasing the rupture area and therefore the released seismic moment (a shear modulus of 30 GPa is used throughout the model computations).

    The epicentres of the generated events are not homogeneously distributed along the fault; being more frequent the generation of events at the tips of the sections and bending (Figure 5a). The rupture initiation in the code is governed by the rate-and-state

formulation. When a nucleation state is reached by an element in the model, the code, spontaneously computes the rupture propagation to the neighbouring elements on a pseudo-dynamic approximation (Dieterich, 1995; Dieterich and Richards-Dinger, 2010; Richards-Dinger and Dieterich, 2012). Towards the ends of the fault is also where the average magnitude is higher (Figure 5c). However, the maximum magnitude of the generated event does not show an important variation; being generated events with magnitudes M > 7.3 along the entire fault trace (Figure 5b). Although the maximum magnitude estimated

previously according to empirical relationships was $M_W$ 7.1 - 7.2 (Álvarez-Gómez et al., 2011a; Gómez de la Peña et al., 2022), our result is close to the maximum magnitude of $M_W$ 7.6±0.3 proposed by Moreno (2011).

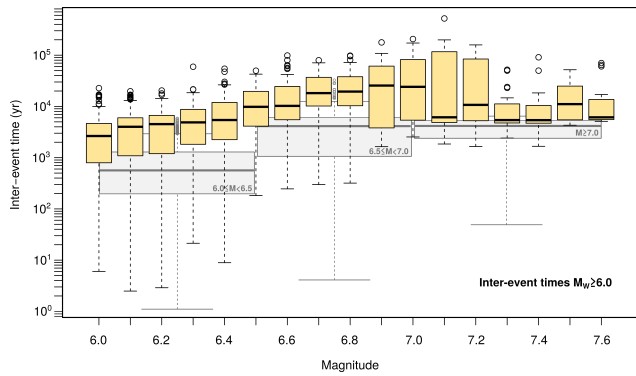

**Figure 4.** Inter-event time distribution. The graph shows a box and whisker plot for intervals of 0.1 in magnitude in yellow. In grey are shown box and whisker plots for different magnitude ranges.

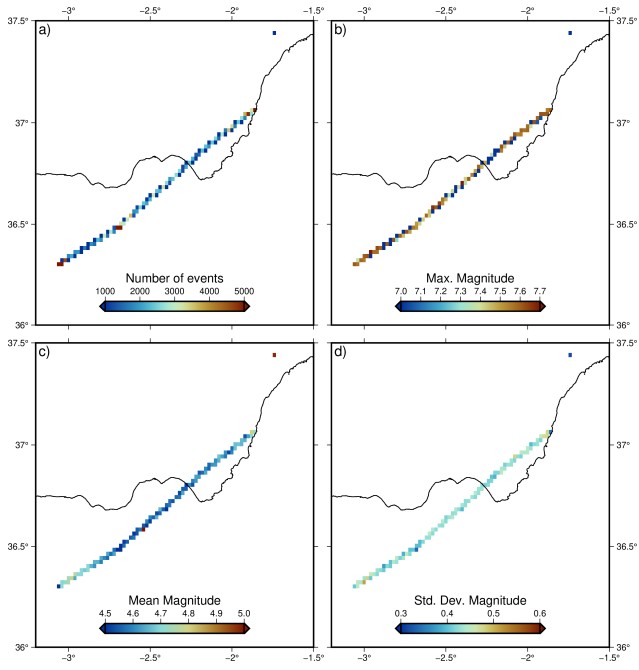

**Figure 5.** Different statistics for the epicentral location of the modelled events shown as heat maps with a cell size of ~5 km². a) Number of epicentres; b) maximum magnitude; c) mean magnitude; d) magnitude standard deviation.

## 3 Tsunami modelling

Simulations for seismic triggered tsunamis are based on modelling the deformation of the ocean bottom produced by the earthquake rupture. These models use analytical solutions in an elastic half-space to reproduce the behaviour of the upper

crust. The most commonly used codes for this are often based on equations derived for rectangular dislocations (Mansinha

and Smylie, 1971; Okada, 1985, 1992), which makes it difficult to model complex rupture geometries without incorporating numerical artefacts. To solve this problem other mathematical approaches and alternative algorithms have been developed, also using analytical equations, but for triangular dislocations (Gimbutas et al., 2012; Meade, 2007; Nikkhoo and Walter, 2015). In this work we have used the calculation algorithm developed by Nikkhoo and Walter (2015) for artefact-free triangular geometries with a Poisson ratio of 0.25, typical of upper crust.

As the faults involved in our models are fundamentally strike-slip structures, we have to take into account the impact of the horizontal component of the deformation in the sea bottom alteration. We have adopted the Tanioka and Satake (1996) approximation, which considers the alteration of the sea bottom morphology by the horizontal displacement of the bathymetric slopes:

$$-u_h = u_x \frac{\partial H}{\partial x} + u_y \frac{\partial H}{\partial y} \tag{2}$$

where $H$ is the water depth and $u_x$ and $u_y$ are the components of the horizontal displacement. The total vertical displacement, applied as initial condition for tsunami generation, is the sum of the vertical component $u_z$ and the horizontal sea bottom alteration $u_h$.

To evaluate the potential of tsunami generation of the modelled earthquakes, we have initially selected events with magnitudes greater than 6.0; obtaining a total of 1,344 events. Many of these events will not have the capacity to generate detectable tsunamis on the coast, so to avoid an excessive computational load, we have filtered these pre-selected events based on the surface deformation generated. As the sea-floor deformation generated by the earthquake is the physical cause producing the tsunami we can use it to directly estimate the tsunamigenic capacity of the event.

Each earthquake rupture is characterized by its unique finite fault model composed by a number of triangular elements. The smaller events considered here, with magnitudes 6.0, are formed by a few tens of elements ($\sim$40); while the biggest ones, with magnitudes of $\sim$7.6 are formed by the rupture of a few thousands of elements (up to 5,279). In total we have modelled the rupture of 1,150,265 triangular elements for the 1,344 finite fault models.

We have parametrized each sea-floor deformation modelled with the following quantities (Bolshakova and Nosov, 2011; Wessel, 1998) (Figure 6):

i) maximum uplift or water elevation

$$\eta_{max} = \max[\eta_Z(x,y)] \tag{3}$$

ii) maximum vertical displacement double-amplitude defined as

$$A_\eta = \max[\eta_Z(x,y)] - \min[\eta_Z(x,y)] \tag{4}$$

iii) displaced volume

$$V = \iint\limits_S |\eta_Z(x,y)| \, dS \tag{5}$$

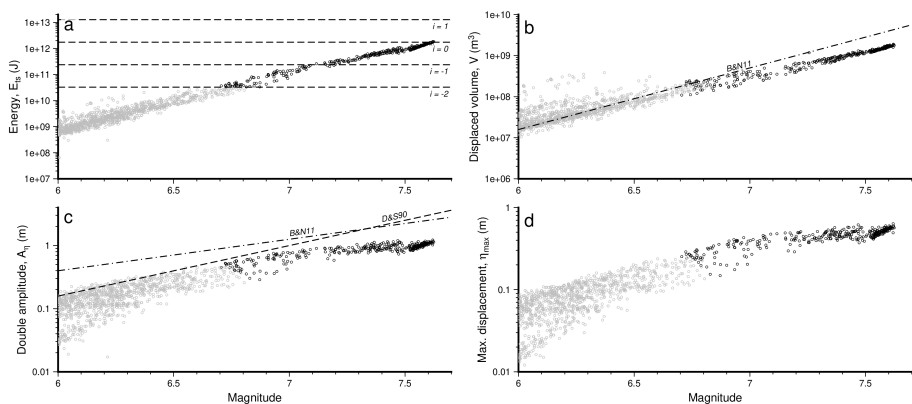

**Figure 6.** Relations of seafloor deformation parameters with earthquake magnitude. a) Potential energy. Dashed lines show the tsunami intensity according to equation 7. b) Displaced volume. Dash-dotted line shows the Bolshakova and Nosov (2011) upper limit for the magnitude - volume relation. c) Displacement double amplitude. Dash-dotted line shows the Bolshakova and Nosov (2011) upper limit for the magnitude - double amplitude relation and the dashed line the Dotsenko and Soloviev (1990) empirical relation. d) Maximum vertical displacement.

and iv) potential energy

$$E_{ts} = \frac{1}{2}\rho g \iint\limits_{S} \eta_Z^2(x,y)dS \tag{6}$$

where $\rho$ is the density of water (taken as 1,038 kg/m$^3$ (Borghini et al., 2014)) and $g$ the acceleration due to gravity.

Nosov et al. (2014) analysed a series of tsunamis generated by earthquakes whose source were characterized with a finite

fault model. They compared the modelled surface deformation with the size and intensity of the generated tsunami. Based on these data, they established a series of relationships between the Soloviev-Imamura intensity of the tsunami (Gusiakov, 2011) and different parameters of the sea-floor deformation, among them the displaced volume and the potential energy. Figure 6a shows the ranges of intensity values defined as a function of the potential energy:

$$i = 1.16\log_{10}(E_{ts}) - 14.2 \tag{7}$$

We have selected to simulate those events with a potential energy capable of generating a tsunami of intensity of at least -2. This criteria restricts the number of tsunami propagations to model from 1,344 events with M > 6 to 331 events with earthquake magnitudes ranging from 6.71 to 7.62 and double amplitudes A$_\eta$ from 0.3 m to 1.2 m.

Bolshakova and Nosov (2011) examined some relevant tsunamis for which they also parametrized the sea-floor deformations. It is noteworthy that those events with double amplitudes below 0.4 m were only perceptible in tide gauges, not generating a

notable impact on the coast. As can be seen in Figure 6 there is a good correspondence between the events selected to simulate with those whose double amplitudes are above 0.3 - 0.4 m.

In order to model the tsunami propagation and inundation we have resort to the highly used and validated code COMCOT (Cornell Multi-grid Coupled Tsunami) (Liu et al., 1995; Wang and Liu, 2006). This algorithm is based on the Non-linear

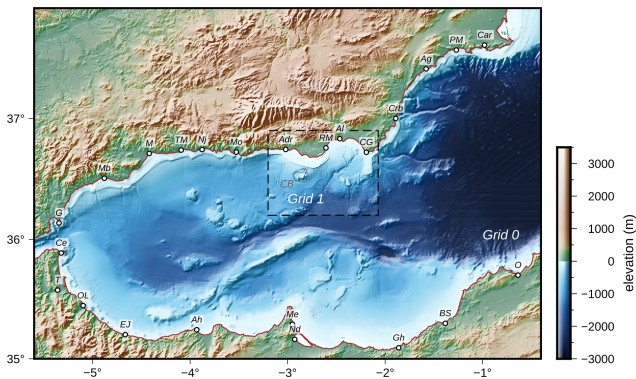

**Figure 7.** Bathymetric grids used in the propagation modelling. The Grid 0, with a cell size of 500m is composed by the EMODnet 2020 bathymetry (EMODnet, 2022) and the topography by the MERIT DEM (Yamazaki et al., 2017). The Grid 1, with a cell size of 100m, is composed by the EMODnet bathymetry and the 25 m resolution topographic DEM of the IGN (CNIG, 2022). The dashed polygon labelled with CB marks the location of the Chella Bank bathymetric feature. The labels show the main localities mentioned in the text: Car, Cartagena; PM, Puerto de Mazarron; Ag, Aguilas; Crb, Carboneras; CG, Cabo de Gata; Al, Almeria; RM, Roquetas de Mar; Adr, Adra; Mo, Motril; Nj, Nerja; TM, Torre del Mar; M, Malaga; Mb, Marbella; G, Gibraltar; Ce, Ceuta; T, Tetouan; OL, Oued Laou; EJ, El Jebha; Ah, Al Hoceima; Me, Melilla; Nd, Nador; Gh, Ghazaouet; BS, Beni Saf; O, Oran.

Shallow Water Equations built over a modified leap-frog nested grids scheme. In this approximation the compressibility of water is not considered, which could act as a filter when computing the initial sea surface deformation (Lotto and Dunham, 2015), and consequently our approach is conservative.

The bathymetry used is composed of three independent sources (Figure 7). On the one hand, the bathymetric data corresponds to the EMODnet 2020 mesh (EMODnet, 2022), with a horizontal resolution of 1/16' (∼115 m). On the other hand, for the regional topography, we have used the MERIT global DEM (Yamazaki et al., 2017), with a horizontal resolution of 3" (∼90 m). For the highest resolution mesh needed in order to compute inundations, on the coast of Almeria, we have used the topography of the digital model of 25 m from the National Geographic Institute of Spain (CNIG, 2022). The regional mesh has been resampled with a cell size of 500 m and the local one with 100 m. We have used open boundary conditions for the water borders, and a Manning's roughness coefficient of 0.02 when computing the inundation.

For each of the 331 tsunami propagations we have computed the maximum elevation for a model running during 90 minutes, which is enough time for the waves to propagate through the basin and capture the wave reflections. In Figure 8 tsunami travel times are shown as well as examples of the results for three events with different magnitudes.

As expected, for the smaller magnitude events, the location of the rupture, as well as the slip distribution along the fault plane, are the determining factor in the location of the maximum wave elevations (Figure 8a-f). However, for the maximum events (Figures 8g-i), in which slip occurs along the entire fault plane (see Figure 2d), the location of the maximum wave elevations are clearly determined by the morphology of the sea-floor.

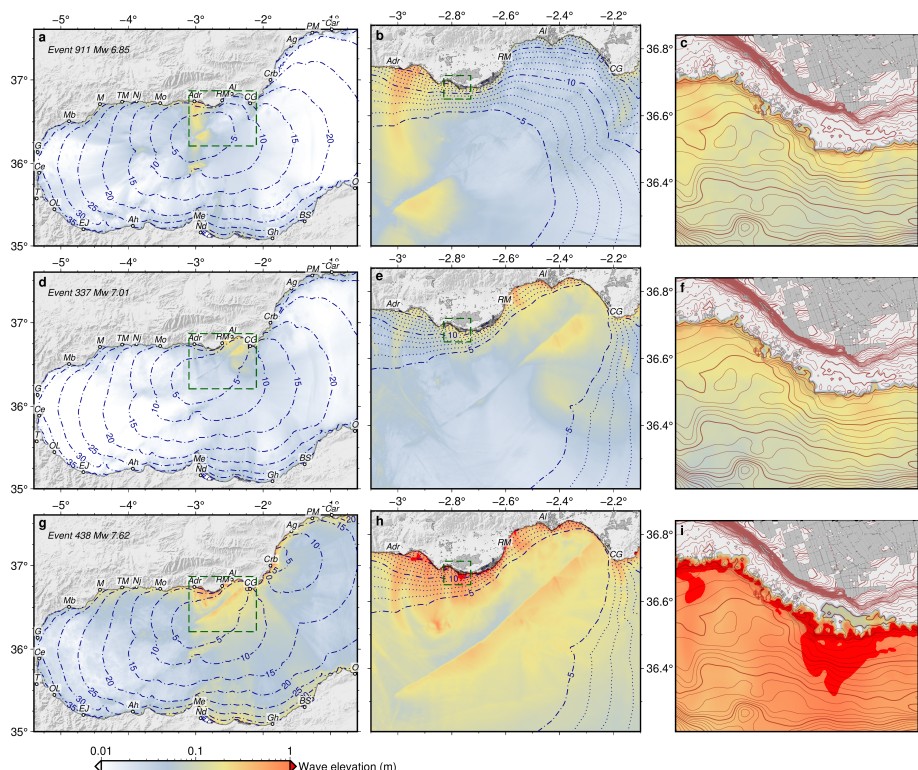

**Figure 8.** Example of maximum wave elevations obtained for three events (out of 331) with different magnitudes. The ruptures corresponding with these events are shown in Figure 2. Dashed contours show the wave travel time in minutes. See Figure 7 for labels of localities. a, d and g show the results for the regional bathymetry; b,e and h show the results in the high resolution topo-bathymetry; c, f and i show examples of the inundation results in the high resolution topo-bathymetry. Green dashed rectangles show the areas of the highest resolution topo-bathymetry as well as the inundation maps.

The classical tsunami hazard deterministic approach consist on the definition of the worst-case scenario based on the dimensions of the source and the employ of a series of empirical relations to define the magnitude of the event and the average slip over the fault. Alternatively, instead of defining a single average homogeneous slip model, a set of stochastic variable slip models can be produced and analysed statistically.

A common procedure is to show the maximum wave elevation expected for each model cell from a series of modelled sources. This kind of map is usually called aggregated maximum elevation map; and is very useful to determine the worst impact of the wave considering all the potential sources. An extension of this reasoning is the aggregation of maximum elevations produced by a set of tsunamis produced by variable slip models on a single source or a set of sources. With this latter approach we have produced the aggregated maximum elevation map shown in Figure 9.

The maximum elevations produced by the Carboneras strike-slip fault exceed 1 m consistently, and with relevant inundations, in the Almerian coast (Figure 9). The maximum elevations are located in front of the fault rupture, from Adra to Almeria city,

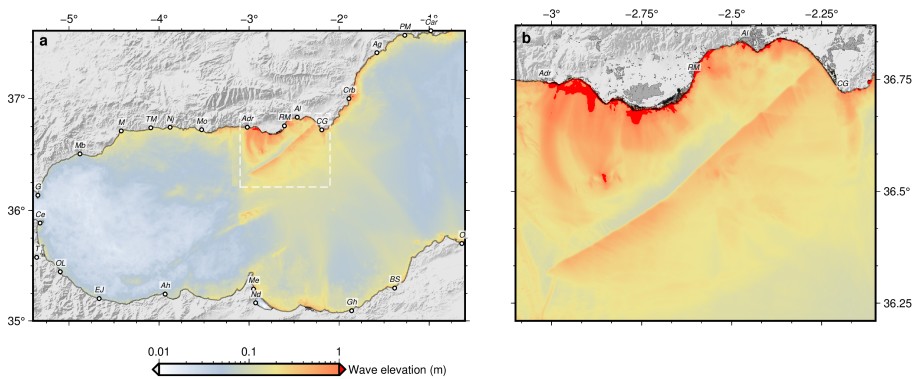

**Figure 9.** Maximum wave elevation aggregated a) regional and b) local maps. See Figure 7 for labels.

but with relevant local inundations in the Cabo de Gata area. Towards the west the maximum elevations can reach locally 1 m but usually show values of a few decimetres. In the opposite coast, in northern Africa, the maximum elevations are always in the range of 0.1 - 0.8 m with the highest values from Melilla to Ghazouet. For north Africa only the coarse bathymetry has been

used and local reflections and resonances not modelled could produce higher elevations locally. Although our approximation is deterministic we can compare our results with the range of elevations obtained for the same area in the context of the probabilistic NEAMTH18 model (Basili et al., 2018, 2021). In the probabilistic model an elevation of 1.3m is obtained with 50% of probability for a period of 10000 years. Our worst-case scenarios, with elevations over 1 m, have inter-event times between 2000 and 6000 years (Figure 4), compatible with the probabilistic hazard results.

Having produced hundreds of rupture scenarios that obey both dimensional and temporal characteristics to the regional seismotectonic context, we can statistically analyse the propagation results. For each calculation point on the map we obtain a statistical distribution of elevations. In Figure 10 some examples are shown for localities along the coast. These elevations have been taken for the 5 m depth isobath. The distribution shown is common for the entire calculation domain, where several local maxima in the distribution can be interpreted, being far from a normal distribution. The highest frequency is commonly

related with the lower elevation values, denoting the lower recurrence interval of small events. The second peak observed is usually related to the highest wave elevations, with values ranging from 1.2 m to 2 m. In some places these higher elevations constitute the most frequent values (as is seen in the example of Figure 10a, corresponding to the locality of Adra). Between these two local maxima in the distribution two more local peaks can be interpreted, although of less importance.

In a simple way we have decided to show the statistical complexity of the distributions of maximum elevations on the coast

through the use of quartiles; thus Figure 11 shows the maximum elevations corresponding to the 25%, 50% (median) and 75% quartiles. The difference between the regional and local maximum elevations arise from the different cell size used in the propagation modelling. The regional bathymetry (grid 0 in Figure 7) has a cell size of 500m, while the local bathymetry (grid 1) has a cell size of 100 m. The coarser bathymetry is unable to reproduce with precision the nearshore complexities and the maximum wave elevations are underestimated. This is clearly shown in the 75% quartile maximum elevations (Figure

11c,f), where the elevations along the coast are consistently higher using the local bathymetry compared with the regional. A

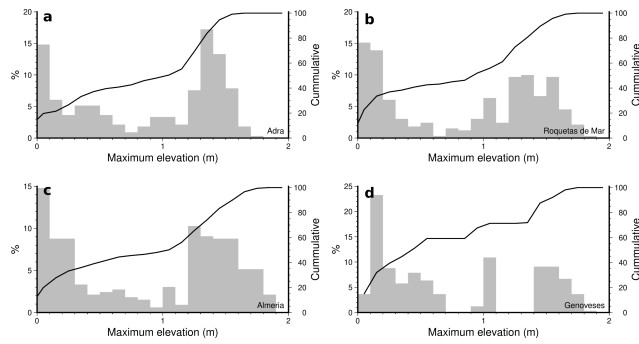

**Figure 10.** Tsunami wave maximum elevations frequency distribution for locations in a) Adra; b) Roquetas de Mar; c) Almería and d) Genoveses Cove.

specially illustrative example is shown in the area of Cabo de Gata (near the eastern edge of the local grid); where a peak on the elevation stands out. This peak is located in the Genoveses cove (Figures 11f) where resonance effects are probably responsible of exceptionally high elevations.

## 4    Discussion

**Tsunamigenic potential of the Carboneras Fault**

In line with results obtained in previous analyses of the tsunamigenic potential of strike-slip faults (Elbanna et al., 2021; Frucht et al., 2019; Gusman et al., 2017; Heidarzadeh et al., 2017; Ho et al., 2021); this work demonstrates the tsunamigenic capacity of the Carboneras Fault. This is a strike-slip fault, with some dip-slip component (rake $\sim$ 10º based on field analysis of its outcrops onshore according to Moreno et al. (2015)) and with the capacity to generate locally damaging tsunamis. However, on a regional scale considering the Alboran Sea basin, its tsunamigenic capacity is more limited, being able to produce tsunamis of small entity in the North African coast between Melilla and Ghazaouet (Figure 9) although with a low frequency (Figure 11).

If we compare the results of this work with previous results (Álvarez-Gómez et al., 2011a, b; Gómez de la Peña et al., 2022) we can see that the tsunamigenic capacity modelled here is higher. While the fault geometry is essentially the same with minor variations due to the higher resolution used in our models than in those of Álvarez-Gómez et al. (2011a) and Gómez de la Peña et al. (2022), there are other parameters that differ significantly. The maximum magnitude, estimated according to empirical relationships, in previous models was $M_W$ 7.1 - 7.2, notably lower than the maximum magnitude reached with our physical model, $M_W$ 7.62, which is close to the maximum magnitude proposed by Moreno (2011). This difference in magnitude consequently produces an important difference in net slip. The one used by Álvarez-Gómez et al. (2011a) is 1.9 m and 1.38 m by Gómez de la Peña et al. (2022); these being average slips over the entire rupture area. In the models that we have developed in this work, the slip is variable, but the average slip for a worst-case of magnitude 7.62 would be $\sim$6 m, with

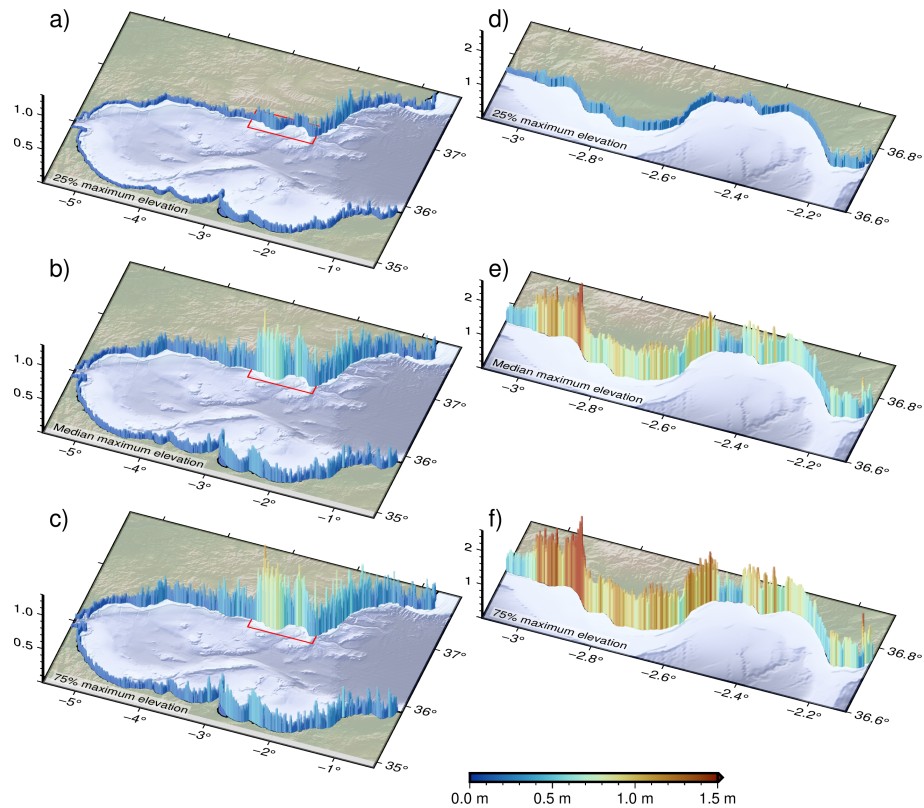

**Figure 11.** Quartiles of maximum wave elevations obtained from the set of modelled sources. Regional scale maps for a) 25% quartile; b) 50% quartile and c) 75% quartile. Local scale maps for d) 25% quartile; e) 50% quartile and f) 75% quartile.

maximum slips ∼9 m. On the other hand, the rake used in our models is 10º while Álvarez-Gómez et al. (2011a) used 15º and Gómez de la Peña et al. (2022) used 0º. Obviously this value has impact on the final results and higher rake values will produce greater wave elevations.

Although the physics-based earthquake simulators generate different catalogues each time they are run, if the boundary conditions are the same, the general picture would be equivalent. For example the range in maximum magnitudes would be probably the same as it depends on the dimensions of the structures. If we vary the slip rate or the seismic coupling, the inter-event times would be different, and consequently the statistical distribution of maximum elevations and inundations. When dealing with deterministic approaches, usually a single worst case scenario is used. In the best case a set of worst case scenar-
ios considering the uncertainties is done. With these models we can obtain a physically coherent seismic behaviour through hundreds of seismic cycles that produce hundreds or thousands of destructive events that can be modelled to characterize not only the worst-case scenario, but also scenarios linked to probabilities of exceedance or return periods.

## Seismogenic potential and Frequency-Magnitude distribution

Although the difference in maximum magnitudes may seem large, it must be taken into account that those provided by the empirical relations are the mean values of the regressions best fits, with standard deviations that may be high. On the other hand, in our models we have selected the largest magnitude generated throughout a 1 Myr. catalogue; and not the average value of the maximum magnitudes generated. If we look at Figure 3, we see that the maximum magnitudes, generated by the complete rupture of the Carboneras fault, vary roughly between magnitudes 6.9 and 7.7 (maximum absolute value of 7.62). If we use the empirical relationship of Leonard (2014) for example, for a maximum rupture length of 71 km (using the Gómez de la Peña et al. (2022) value) we obtain magnitudes of 7.25 and using the rupture area we obtain values of 7.14. However, these values represent the mean of the best fit, with a one standard deviation range between 6.86 and 7.64 for the empirical length-$M_W$ relationship and between 6.88 and 7.4 for the area-$M_W$ relationship. Therefore, the values obtained in our model are within the range of one standard deviation of this empirical relationship, with the advantage that we can obtain an in-depth maximum magnitudes statistical analysis.

The distribution of frequencies and magnitudes (FMD) of the generated seismic catalogue (Figure 3) reflects the variability of magnitudes associated to the rupture of the entire fault. Since the long-term behaviour of the modelled system is complex, although the construction of the model is deterministic, the statistical distribution of the generated events reflects the stochastic behaviour characteristic of dynamical systems showing self-organized criticality (SOC) (Bak et al., 1988; Bak and Tang, 1989). This stochastic behaviour of the system is reflected also in the non-linearity of the relationship between the size of the earthquake rupture and the slip; thus, for the same rupture size we obtain different slip distributions and therefore different magnitudes are generated. Maybe as a consequence of this, the upper limit of the FMD departs from the classical Gutenberg-Richter potential relation (GR), showing a distribution markedly different that can be seen on the discrete counts of the plot in Figure 3. This distribution resembles that of the characteristic earthquake behaviour (Schwartz and Coppersmith, 1984), showing a bell-shaped distribution of the characteristic earthquake magnitudes. Similar results have been observed also in other physics-based models (Console et al., 2021; Rafiei et al., 2022; Shaw et al., 2022). In our models this behaviour can be related to the physical limit imposed to the maximum rupture area and consequently limiting the self-similar range of the dynamic system (Ben-Zion and Rice, 1995).

## Worst-case scenario approximation

This concept of characteristic earthquake, or worst-case scenario, is frequently used in deterministic hazard approximations. In these models, simple rectangular sources with homogeneous slips over the fault rupture are used. There has been much debate about the appropriateness of using these simple models and whether they can roughly reflect the tsunamigenic potential of a source. To address some of the drawbacks of this methodology, stochastic probabilistic approaches have been proposed for the generation of variable slips on the fault plane. In principle, the variable slip should play a key role on the impact of local sources, which has been seen in the models shown in Figure 8 for events of different magnitudes (see supplementary models for comparison). To analyse the influence of these aspects we have compared one of the maximum scenarios modelled, with

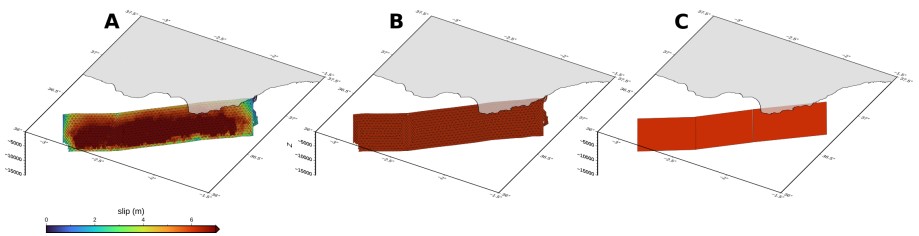

**Figure 12.** Comparison of source simplifications. a) Realistic geometry and variable slip maximum earthquake, $M_W$ 7.62, model; b) realistic geometry with average homogeneous slip; c) simplified geometry and homogeneous slip.

magnitude $M_W$ 7.62 (Figure 12a), with scenarios of equal magnitude but with homogeneous slip in the detailed geometry (Figure 12b), as well as with a simplistic model based on 3 rectangular sections like the one used by Álvarez-Gómez et al. (2011a) (Figure 12c).

At first glance, the propagations are very similar and share their main features. If we compare them at a local scale (Figure 13), we see that the differences are below 0.5 m in general, although locally the differences may be greater on the coast (up to 3 m). From the regional point of view, the differences are minor (Figure 14).

If we compare the local propagation between the simplified rectangular source and the variable slip source (Figure 13d) the main differences are located towards the tips of the fault sections. These sections behave as patches whose slip decreases towards the tips (Figure 12a), and it is therefore at these points where the simplified model overestimates sea-floor uplift (the blue colours in figure 13d show this important difference in the Adra area). On the contrary, towards the centre of the sections the simplified model underestimates the uplift. These differences are essentially the same as those that can be observed in the comparison of both realistic geometries but with variable or constant slips (Figure 13e). In this case, since both geometries are the same, the differences between both models are minor. Some of these differences can be diminished using slip tapering towards the edges of the rectangular rupture or modelling elliptical ruptures.

Regionally, the differences between the models are minor, although the slip differences towards the southern tip of the fault are evident, as has been seen locally as well. On both the Iberian and African coasts, values are overestimated by the simplified models towards the western part of the basin (negative values in Figures 14d and e), while elevations are underestimated towards the eastern part.

What is evident is the main role that bathymetry plays in the propagation features, determining to a large extent the location of the areas where there is major impact (Figure 9). In this sense, the Chella Bank, off the coast of Adra (Figure 7), determines the wave propagation and the impact on this coast, in which the highest wave elevations are observed (Figure 11).

## 5   Conclusions

From a deterministic point of view, the one adopted in this work, the use of physics-based earthquake simulations for tsunamigenic sources allows an in-depth characterization of the scenarios, either through aggregated maps of maximum elevations

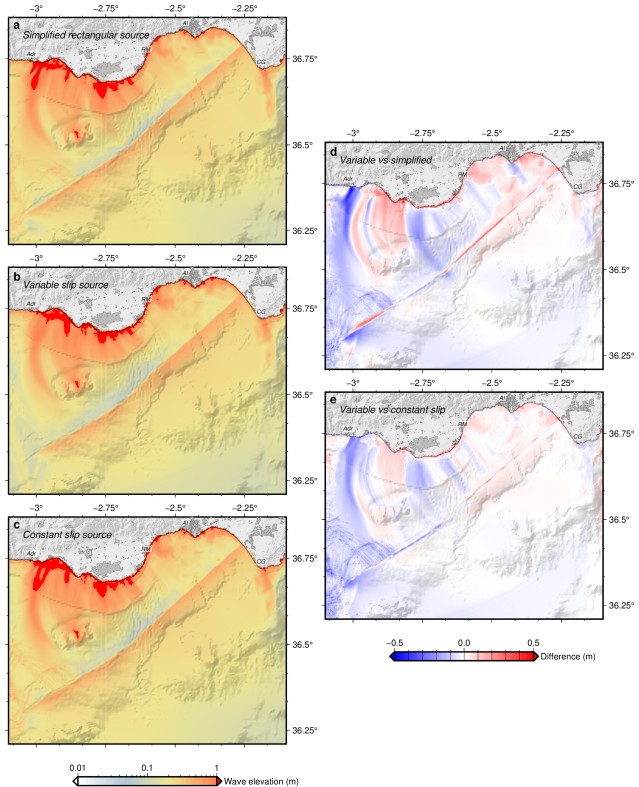

**Figure 13.** Comparison of local propagations for the simplified sources shown in Figure 12. a) Propagation of the simplified rectangular source. b) Propagation of the variable slip and realistic geometry source. c) Propagation of the homogeneous slip source with realistic geometry. d) Differences between variable and simplified sources. e) Differences between variable and constant slip sources. See Figure 7 for labels.

(Figure 9) or the statistical exploitation of the hundreds or thousands of scenarios generated (Figure 11). In addition, the use of this tool allows characterizing the inter-event times and the recurrence intervals of the maximum events, which are those that have the greatest impact on tsunami hazard.

Regarding the estimation of the maximum magnitude of a source, a key step in the deterministic characterization of the tsunami impact, this methodology incorporates the stochastic natural variation in rupture area, slip and magnitude that arises

from the non-linear process of seismic rupture. Thus, instead of characterizing the size of the worst-case earthquake through empirical relationships, we can obtain a range of magnitudes characterized by a probability distribution, which allows an in-depth implementation of uncertainty estimation. In addition, each modelled seismic rupture is characterized by its own variable slip distribution and rupture process as they are modelled with a quasi-dynamic algorithm.

The strike-slip Carboneras Fault has the capacity to generate locally damaging tsunamis. However, on a regional scale,

considering the Alboran Sea basin, its tsunamigenic capacity is more limited. Comparing our results with previous works

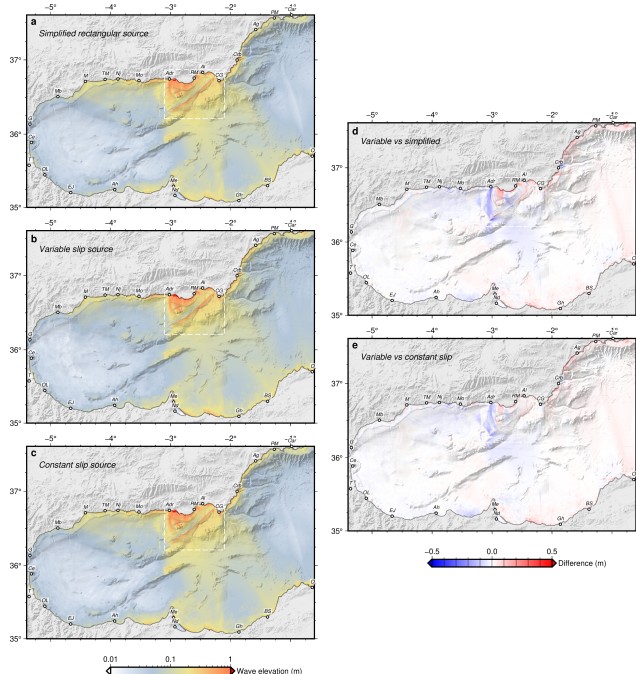

**Figure 14.** Comparison of regional propagations for the simplified sources shown in Figure 12. a) Propagation of the simplified rectangular source. b) Propagation of the variable slip and realistic geometry source. c) Propagation of the homogeneous slip source with realistic geometry. d) Differences between variable and simplified sources. e) Differences between variable and constant slip sources. See Figure 7 for labels.

(Álvarez-Gómez et al., 2011a, b; Gómez de la Peña et al., 2022) we can see that the tsunamigenic capacity modelled here is higher, basically due to the difference in maximum magnitude, which produces an important difference in maximum net slip.

The distribution of frequencies and magnitudes (FMD) of the generated seismic catalogue (Figure 3) reflects the variability of magnitudes associated to the rupture of the entire fault. The upper limit of the FMD departs from the classical Gutenberg-
385 Richter potential relation, showing a bell-shaped distribution of the maximum earthquakes magnitude in a range between 6.9 and 7.7. The inter-event time for these magnitudes is around 2000 – 6000 years (Figure 4).

The use of physics-based earthquake simulations for tsunamigenic sources allows a qualitative leap in their characterization. From a probabilistic point of view, these models have shown in the Probabilistic Seismic Hazard Analyses (PSHA) a great potential to estimate recurrence periods and inter-event times for large earthquakes, which are poorly represented in the in-
390 strumental seismic catalogues (Chartier et al., 2021; Herrero-Barbero et al., 2021; Console et al., 2017); being one of the key pieces in the current development of seismic forecast models (Dieterich and Richards-Dinger, 2010; Field, 2019; Field et al., 2014; Shaw et al., 2018). The implementation of these methodologies in the Probabilistic Tsunami Hazard Analyses (PTHA), is a logical and necessary step.

*Code availability.* The RSQSim physics-based earthquake simulator has been developed by Richards-Dinger and Dieterich (2012) and can be obtained from the authors upon request. The COMCOT tsunami simulation has been developed by Liu et al. (1995). The version used in this work is an adaptation to gfortran compiler developed by Tao, Chiu at Tsunami reseach group, IHOS, NCU; available at https://github.com/AndybnACT/comcot-gfortran. GMT (Wessel et al., 2013) has been used to perform some calculations and to produce most of the figures.

*Author contributions.* JAAG conceived the work, performed the tsunami simulations and analysis and collaborated on the manuscript writting. PHB contructed the 3D fault model, performed the physics-based earthquake simulations and analysis and collaborated on the manuscript writting. JJMD collaborated on the manuscript writting and on the discussions on the EBSZ tectonics.

*Competing interests.* The authors declare no competing interests.

*Acknowledgements.* The bathymetry used has been provided by the EMODnet Bathymetry Consortium (EMODnet, 2022). This work has been partially funded by the project model_SHaKER (PID2021-124155NB-C31) of the Spanish National Research Agency. We would like to thank the comments and constructive criticism of the reviewers J. García-Mayordomo and L. Matías, as well as the useful suggestions of the editor R. Omira.

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
