# Peer review of "Seismogenic potential and tsunami threat of the strike-slip Carboneras Fault in the Western Mediterranean from physics-based earthquake simulations"

_Natural Hazards and Earth System Sciences, 2022_

## Referee Comment (RC1)

**NHESS-2022-186 Seismogenic potential and tsunami threat of the strike-slip Carboneras Fault in theWestern Mediterranean from physics-based earthquake simulations**

José A. Álvarez-Gómez, Paula Herrero-Barbero, and José J. Martínez-Díaz

**Comments by Luis Matias, University of Lisbon**

**Recommendation**

It is my recommendation that the work deserves publication but that it requires major revision. The reasons for this evaluation are detailed below. In fact, my suggestion that may, or may not, be endorsed by the authors is to split the work into two parts. Part 1 dedicated to the generation of the earthquake catalogue and Part 2 dedicated to the deterministic evaluation of tsunami hazard in the area. Further additional comments are provided in another section and are given on the annotated pdf.

**Major comments**

The authors apply an earthquake physical model to generated 1 Myr catalogue of earthquakes along the Carboneras Fault (CBF) and adjacent faults. The model is constrained by the assumed fault slip rates and some parameters are tuned to fit an ad-hoc Gutenberg-Richter law. The physical model generates ruptures with variable slip distribution and this feature is explored on a second part of the paper where tsunamis are generated. To our knowledge it is the first time that such physical models for earthquake generation are used in Iberia and surrounding seismically active domains. Such an effort deserves publication on itself, but additional details, and discussion must be provided to encourage the application of the model in other domains. The additional information to be provided may lead to a growth of the paper that could imply splitting it into two parts, Part 1 dedicated to the generation of the earthquake catalogue and Part 2 dedicated to the deterministic evaluation of tsunami hazard in the area. My comments will also be split according to this suggestion.

Part 1:The physical model for earthquake generation

My major concern regarding this subject is the lack of relationship between observed seismicity and the Carboneras Fault. This can be inferred from Figure 1 in the paper but is made clear on figure REV-01.

[Figure]

*Fig. REV-01. The Carboneras Fault (black line) and recorded seismicity from ISC.*

The paper mentions that some model parameters are tuned so that the final Gutenberg-Richter (GR) law has a b value equal to 1.0. The paper fails to give the support for this assumption and no information is provided on the a value that also characterizes the GR law. Assessing the ISC catalogue and selecting a generous area surrounding the CBF we obtain the GR law shown in figure REV-02, where the number of earthquakes is scaled to 1 Myr as in the paper.

We obtain a very high b-value, not common for convergent or transcurrent domains, showing that large magnitude events are much less frequent than found on average on the earth. This may be a feature due to the small number of events, but it deserves discussion. The thickness of the brittle layer assumed for the physical model deserves additional discussion in the light of information provided by the earthquake catalogues and deep structure studies in the area.

[Figure]

*Fig. REV-02. The Carboneras Fault (black line) and recorded seismicity from ISC.*

If we compare the compute GR law with the earthquake distribution published, as shown in figure REV-03, we remark that, as suspected, the observed seismicity is much less than the modelled catalogue. This feature deserves to be discussed in the paper.

[Figure]

*Fig. REV-03. Comparison of figure REV-02 and the paper's figure 3.*

Another constrain on the physical model is the average slip rate on the CBF and neighbouring faults. The authors used for the CBF the value 1.3 ± 0.2 mm/yr proposed by Echeverria et al. (2015). We quote here the Echeverria et al. (2915) sentences (CFZ = Carboneras Fault Zone): "*The analysis of GPS data in the SE Betics confirmand quantify the ongoing tectonic activity of the onshore segment of the CFZ as a left-lateral strike–slip fault. For the first time, we were able to provide a quantitative measure of the present-day horizontal geodetic slip-rate of the CFZ, suggesting a maximum left-lateral motion of 1.3 ± 0.2 mm/yr. The coincidence of geologic and geodetic strike–slip rates along the CFZ, illustrates how during Quaternary its northern segment has been tectonically active and has been slipping at a rate of 1.1 to 1.5 mm/yr*".

It is clearly suggested that 1.3 is a value valid only for the onshore segment of the CBF and it represents the maximum value. The use of this value deserves further discussion as well as the consequences for its variation along the fault, particularly on the ocean segment. Furthermore, our interpretation of Echeverria et al. (2015) figure 5 is that the CBF slip rate, as measured by GPS, lies between 1.0 and 2.5 mm/yr.

It may be relevant here for the authors to mention other sources of information on the fault slip of oceanic faults as provided by Neotectonic modelling. While Jiménez-Munt & Negredo (2003) and Cunha et al. (2012) provide slip rates estimates smaller

than 0.5 mm/yr, Neres et al. (2016) show a maximum value of 1.7 mm/yr for the CBF.

We understand that a perfect coupling is assumed for the CBF and neighbouring faults between the kinematic constrain (slip rate) and the earthquake generation. The typical seismic coupling of major plate boundary types has been discussed by Bird and Kagan (2004). They showed that for continental convergent boundaries it lies between 0.51 and 1.00 (1.00 preferred value) and for continent transform faults it lies between 0.38 and 1.00 (0.72 preferred value). The author's choice of 1.0 deserves some discussion and the consequences of using a different value should be addressed.

It seems that the generation of "characteristic earthquake" recurrence models is a feature of the physical model used. This model has not been used of PSHA and PTHA in Europe and additional discussion should be provided. Is it a model feature or is it explained by some characteristics of the CBF domain?

Another feature of the physical model for earthquake recurrence applied to the CBF system is that the maximum magnitude exceeds the estimations made by several authors. It is argued that the maximum magnitude value lies at the extreme boundary of some estimates. Is it a feature of the model? Why does it happen? Some additional discussion is needed here.

The physics-based earthquake generation model, besides the definition of the geometry (that itself deserves additional support), requires many parameters, some to be defined and others used as constrains. A list of the main parameters and information on its choice must be well presented, which is not the case on the current version of the paper. If possible, several runs of the model could be used, first to assess the random uncertainty and then to assess the epistemic uncertainty due to the choice done for some selected critical parameters. One set of the parameters, or features, of the physics-based earthquake generation model that is not explained at all, and is relevant for the higher magnitude events, is the one that rules the multi-segment propagation.

Part 2:Tsunami

Here the major comment regards the absence of any reference to previous works on tsunami hazard assessment in the area. The major study requires some reference and comparison is the NEAMTH18 (Basili et al., 2019, 2021). The results of this study can be assessed online[1]. The area investigated by the paper is shown in figure REV-04 as well as the probability results for the Almeria forecast point.
* * *
[1] http://ai2lab.org/tsumapsneam/interactive-hazard-curve-tool/

[Figure]

*Fig. REV-04. NEAMTH18 forecast points for the Western Mediterranean and Hazard results for Almeria.*

**Additional comments**

On the pdf provided most of the figures are small and difficult to read.

*Lines 21-22: this ability depends on their mode of seismic rupture*

In fact, this is not the major parameter defining the potential for tsunami generation by an earthquake, as can be verified by the decision matrix adopted for the ICG/NEAMTWS (2011). The main parameters are:

- Top of the fault depth (focal depth if that information is not available)
- Location in relation to the coastline
- magnitude

Given the high directivity of tsunami propagation we may add also the strike of the structure. We suggest the authors to frame better the above-mentioned sentence.

*Line 27: Although the lower capacity of strike-slip faults to generate tsunamis is a proven fact,*

Mention here the strike-slip generated tsunamis in the Gloria Fault, one domain close to the one investigated in the paper and both belonging to the Nubia-Eurasia plate boundary, as presented by Baptista and Miranda (2009).

*Line 33: based on the simulation of tsunamis generated by ruptures of simple, rectangular*

This is one occasion to mention the NEAMTHM18 model (Basili et al., 2019, 2021) that covers the investigated area and didn't use simple rectangular sources.

*Line 43:*

Add to the list the NEAMTHM18 model (Basili et al., 2019, 2021)

*Figure 1*

The location of Almeria is missing, and it is needed.

The geographic projection is "Plate Carrée" which is very unusual. Renders the comparison with other maps difficult. Why this projection was used?

*Line 73: this fault has been proposed as source of the 1522 Almeria earthquake*

This sentence is in contradiction with the location of the event in Figure 1. Clarify.

*Line 79: Although the tsunami simulations done to date*

Add to the list the NEAMTHM18 model (Basili et al., 2019, 2021) and comment briefly its methodology.

*Lines 108-109: fault depths between 8 and 12 km.*

It is no clear what the authors mention as "fault depth". Is it the width of the fault? On which data is this information based?

*Figure 2*

Explain in the caption the shorthand terms used, e.g., ASMF, PF, …

*Line 115: Besides the input kinematic data*

Our interpretation is that the authors model the ruptures on the CF and also on other faults to the NE. What is the slip rate on these faults? What was the source of information?

*Line 118: We define reference rate-and-state values based on experimental data*

This experimental data was likely sampled at shallow depths. How do they apply to the expected seismogenic depths? If they are assumed to be identical on the whole fault system, what might be the consequences of this simplifying hypothesis?

*Line 122: rate-and-state friction parameters a=0.001 and b=0.010;*

There might be a confusion with the a and b parameters of the Gutenberg-Richter law.

*Line 125: also suggested possible creeping sections in the Carboneras fault*

How would this hypothesis affect the paper results? Some discussion on the assumptions and simplifications made should be provided. See main comment earlier in this document.

*Figure 3*

Given that the physics-based earthquake generation model is applied to a set of faults, it is not clear if the histogram applies only to CF or to the whole system as depicted in Figure 2. It is assumed that the a priori average slip rate of the CF is respected by the model, but a sentence presenting the a posteriori compute slip rate is needed.

*Line 132: excluding the aftershocks*

Since it seems that the authors are discussing the 6.5 to 7.0 magnitude interval, what is the definition of aftershocks used?

*Line 137: therefore the released seismic moment.*

What is the shear modulus value used to compute seismic moment? Justification for that value? Is it uniform along the fault and over depth? What are the consequences of using a single average value for the modelling?

*Line 138: The epicentres*

How does the code obtain the slip initiation? Is it a feature of the code or the epicentre is computed a posteriori from the rupture distribution?

*Lines 154-155: As the sea-floor deformation generated by the earthquake is usually transferred instantly to the 155 elevation of the water free surface*

This is not true in general, though it applies to the modelling of far source tsunamis. For locally generated tsunamis there are two effects that are not considered in the paper that deserve a comment: i) the finite compressibility acts as a filter when computing the sea surface deformation (e.g. Lotto & Dunham, 2015); ii) the horizontal movement of the sea bottom, in areas of relief, generate an initial velocity on the water that, in some circumstances, must be considered.

*3. Tsunami modelling*

There are a few general questions that must be addressed by the authors.

1) What are the boundary conditions used for the water borders?

2) What are the elastic parameters used to compute the seafloor deformation? Justification?

3) What happens close to the coastline? Is friction used? What are its characteristics?

4) Is there inundation?

5) How is the tsunami amplitude computed? It is recommended that the tsunami wave amplitude to be computed at cells with water depth no smaller than 50 m. The reason is explained in Kamigaichi (2011): "To represent the tsunami waveform correctly in a shallow sea area, very fine bathymetry data mesh is necessary (in a strict sense, 20 or more grid points are necessary within one wave-length [31]), and a vast time is required for the completion of such detailed calculations. To overcome this difficulty, the numerical simulation with the long-wave approximation is applied only to points which are a few to a few ten kilometers seaward from the coast ("forecast points") where sea depth is about 50m. Then, tsunami amplitude at the coast is calculated by using Green's law described in the next section."

*Line 204: maximum wave elevation*

The meaning of this parameter must be well explained. See my previous comment.

*Line 211: relevant local inundations*

It is not explained how "maximum elevation" is converted to inundation. See previous comments.

*Figure 8*

Explain the dashed contours.

*Line 218: have been taken for the 5 m depth isobath*

This explanation should have been provided earlier. Given previous comments 5 m seems not appropriate. What happens if there is no cell at 5 m depth? Use the Green law to convert it to 5 m?

*Lines 224 - 234*

Given that a single, randomly generated, catalogue was used, I fail to see the relevance of the discussion on these details of the tsunami amplitude histogram. Would another catalogue generate the same features?

*Line 239: generate locally damaging tsunamis*

Define "local" tsunamis. The term "local" has a very specific meaning in the tsunami warning systems (ICG/NEAMTWS ,2011)

*Line 243: If we compare the results of this work with previous results*

Given that the "frequency" of tsunamis was mentioned in line 241, comments on the NEAMTHM18 model (Basili et al., 2019, 2021) are appropriate here.

*Line 248: Mw 7.62, which is close to the maximum magnitude proposed by Moreno (2011).*

This value and reference were not mentioned in the introduction and they are relevant.

*Lines 252 – 253: On the other hand, the rake used in our models is 10° while Alvarez-Gomez et al. (201 la) used 15º and G6mez de la Pefia et al. (2022) used 0°.*

What are the consequences of the uncertainty on the rake to the paper results?

*Lines 264 – 265: we can obtain the maximum magnitudes in a robust manner from a statistical point of view.*

Given the larger number of simplifications and approximations used in the physics-based earthquake generation model, given that a single catalogue was generated without assessing aleatoric and epistemic uncertainties, I cannot classify the results as "robust", though deserving to be published.

*Lines 270 – 271: relationship between the size of the earthquake rupture and the slip;*

Not shown in the paper. Show as supplement?

*Line 278: would reflect a GR relation*

In fact, one of the most frequent earthquake recurrence laws is the truncated GR relation, not the simple (and open) GR law.

*Line 285: see supplementary models*

These are not available on the documentation provided.

*Line 295: the simplified model overestimates*

It should be mentioned that nowadays the common procedure is to taper the uniform slip at the borders of the rectangular faults (e.g. Davies and Griffin, 2018). Given this, the comparison between tsunamis generated by irregular and uniform slip faults is unfair, for the tips of the fault as mentioned in the text.

*Figure 13*

The labels mentioned in the caption cannot be seen on the figure. Too small?

*Line 309: allows a more robust characterization of the scenarios*

Given the larger number of simplifications and approximations used in the physics-based earthquake generation model, given that a single catalogue was generated without assessing aleatoric and epistemic uncertainties, I cannot classify the results as "robust", though deserving to be published.

*Figure 14*

The labels mentioned in the caption cannot be seen on the figure. Too small?

*Lines 316, 317: which allows a robust implementation of uncertainty estimation*

Given the larger number of simplifications and approximations used in the physics-based earthquake generation model, given that a single catalogue was generated without assessing aleatoric and epistemic uncertainties, I cannot classify the results as "robust", though deserving to be published.

*Lines 332, 333: The implementation of these methodologies in the Probabilistic Tsunami Hazard Analyses (PTHA) is a logical and necessary step.*

Comment/discussion of the NEAMTHM18 model (Basili et al., 2019, 2021) are needed here since they apply to the same area discussed in the paper.

*Line 337: GMT (Wessel et al., 2013) has been used to*

Given this information we do not understand the use of the "Plate Carrée" projection in Figure 1.

**References mentioned in the comment, not cited in the paper**

Baptista, M. A., & Miranda, J. M. (2009). Revision of the Portuguese catalog of tsunamis. Natural hazards and earth system sciences, 9(1), 25-42.

Baptista, M. A., Miranda, J. M., Matias, L., & Omira, R. (2017). Synthetic tsunami waveform catalogs with kinematic constraints. Natural Hazards and Earth System Sciences, 17(7), 1253-1265.

Basili, R., Brizuela, B., Herrero, A., Iqbal, S., Lorito, S., Maesano, F. E., ... & Zaytsev, A. (2021). The making of the NEAM tsunami hazard model 2018 (NEAMTHM18). Frontiers in Earth Science, 753.

Basili, R., Brizuela, B., Herrero, A., Iqbal, S., Lorito, S., Maesano, F. E., ... & Oueslati, F. (2019). NEAMTHM18 documentation: the making of the TSUMAPS-NEAM tsunami hazard model 2018.

Bird, P., & Kagan, Y. Y. (2004). Plate-tectonic analysis of shallow seismicity: Apparent boundary width, beta, corner magnitude, coupled lithosphere thickness, and coupling in seven tectonic settings. Bulletin of the Seismological Society of America, 94(6), 2380-2399.

Cunha, T. A., Matias, L. M., Terrinha, P., Negredo, A. M., Rosas, F., Fernandes, R. M. S., & Pinheiro, L. M. (2012). Neotectonics of the SW Iberia margin, Gulf of Cadiz and Alboran Sea: a reassessment including recent structural, seismic and geodetic data. Geophysical Journal International, 188(3), 850-872.

Davies, G., & Griffin, J. (2018). The 2018 Australian probabilistic tsunami hazard assessment: Hazard from earthquake generated tsunamis. Geoscience Australia.

ICG/NEAMTWS (2011). Interim Operational Users Guide for the Tsunami Early Warning and Mitigation System in the North-eastern Atlantic, the Mediterranean and Connected Seas (NEAMTWS) Version 2.00 Approved by ICG/NEAMTWS-VIII (Santander, 22-24 November 2011). Available online at https://www.ingv.it/cat/images/images/documenti/NEAMTWS-OpUG-version_2_0_Final.pdf

Jiménez-Munt, I., & Negredo, A. M. (2003). Neotectonic modelling of the western part of the Africa–Eurasia plate boundary: from the Mid-Atlantic ridge to Algeria. Earth and Planetary Science Letters, 205(3-4), 257-271.

Kamigaichi, O. "Tsunami forecasting and warning. Extreme environmental events complexity in forecasting and early warning." (2011): 982-1007.

Lotto, G. C., & Dunham, E. M. (2015). High-order finite difference modeling of tsunami generation in a compressible ocean from offshore earthquakes. Computational Geosciences, 19(2), 327-340.

Neres, M., Carafa, M. M. C., Fernandes, R. M. S., Matias, L., Duarte, J. C., Barba, S., & Terrinha, P. (2016). Lithospheric deformation in the Africa-Iberia plate boundary: Improved neotectonic modeling testing a basal-driven Alboran plate. Journal of Geophysical Research: Solid Earth, 121(9), 6566-6596.

---

## Referee Comment (RC2)

[referee-annotated manuscript omitted]

---

## Author Comment (AC3)

[revised manuscript text omitted]

(1)

where $a$ and $b$ are the dimensionless rate-and-state coefficients; $V$ and $V_0$ are the slip speed and its normalizing value; $D_C$ is the characteristic distance for slip evolution (10-5 m in this study); and $\mu_0$ is a steady-state coefficient that evolves with slip and normal stress changes, deriving in consequent equations that can be consulted in Richards-Dinger and Dieterich (2012) with more detail.

Stress interactions are applied on each fault element following Okada (1992) elastic dislocation solutions:

$$\dot{\tau}_i = K_{ij}^{\tau} V_j + \dot{\tau}_i^{tect},$$
(2)

and

$$\dot{\sigma}_i = K_{ij}^{\sigma} V_j + \dot{\sigma}_i^{tect},$$
(3)

where $\tau_i$ and $\sigma_i$ are the shear stress in the direction of slip and normal stress on the i-th element, respectively; $\dot{\tau}_i^{tect}$ and $\dot{\sigma}_i^{tect}$ are constant stressing rates applied to the i-th element by sources external to the fault system (equivalent to far-field tectonic stress); and $K^{\tau}$ and $K^{\sigma}$ are the stiffness matrices. To achieve 
[revised manuscript text omitted]
 dot located on the upper-right corner of the map shows an event nucleated on the Palomares Fault but rupturing also the Carboneras fault.**

195     To evaluate the potential of tsunami generation of the modelled earthquakes, we have initially selected events with magnitudes greater than 6.0; obtaining a total of 1344 events. Many of these events will not have the capacity to generate detectable tsunamis on the coast, so to avoid an excessive computational load, we have filtered these pre-selected events based on the surface deformation generated. **As the sea-floor deformation generated by the earthquake is the physical cause producing the tsunami we can use it to directly estimate the tsunamigenic capacity of the event.**

200     Each earthquake rupture is characterized by its unique finite fault model composed by a number of triangular elements. The smaller events considered here, with magnitudes 6.0, are formed by a few tens of elements ($\sim$40); while the biggest ones, with magnitudes of $\sim$7.6 are formed by the rupture of a few thousands of elements (up to 5279). In total we have modelled the rupture of **1,150,265** triangular elements for the 1344 finite fault models.

    We have parametrized each sea-floor deformation modelled with the following quantities (Bolshakova and Nosov, 2011;
205  Wessel, 1998) (Figure 6):

i) maximum uplift or water elevation

$$\eta_{max} = \max[\eta_Z(x,y)] \tag{4}$$

ii) maximum vertical displacement double-amplitude defined as

$$A_\eta = \max[\eta_Z(x,y)] - \min[\eta_Z(x,y)] \tag{5}$$

[Figure]

**Figure 6.** Relations of seafloor deformation parameters with earthquake magnitude. a) Potential energy. Dashed lines show the tsunami intensity according to equation 8. b) Displaced volume. Dash-dotted line shows the Bolshakova and Nosov (2011) upper limit for the magnitude - volume relation. c) Displacement double amplitude. Dash-dotted line shows the Bolshakova and Nosov (2011) upper limit for the magnitude - double amplitude relation and the dashed line the Dotsenko and Soloviev (1990) empirical relation. d) Maximum vertical displacement.

210    iii) displaced volume

$$V = \iint\limits_{S} |\eta_Z(x,y)| dS \tag{6}$$

and iv) potential energy

$$E_{ts} = \frac{1}{2} \rho g \iint\limits_{S} \eta_Z^2(x,y) dS \tag{7}$$

where $\rho$ is the density of water (taken as 1038 kg/m³ (Borghini et al., 2014)) and $g$ the acceleration due to gravity.

215    Nosov et al. (2014) analysed a series of tsunamis generated by earthquakes whose source were characterized with a finite fault model. They compared the modelled surface deformation with the size and intensity of the generated tsunami. Based on these data, they established a series of relationships between the Soloviev-Imamura intensity of the tsunami (Gusiakov, 2011) and different parameters of the sea-floor deformation, among them the displaced volume and the potential energy. Figure 6a shows the ranges of intensity values defined as a function of the potential energy:

220    $i = 1.16 \log_{10}(E_{ts}) - 14.2$ \hfill (8)

We have selected to simulate those events with a potential energy capable of generating a tsunami of intensity of at least -2. This criteria restricts the number of tsunami propagations to model from 1344 events with M > 6 to 331 events with earthquake magnitudes ranging from 6.71 to 7.62 and double amplitudes $A_\eta$ from 0.3 m to 1.2 m.

Bolshakova and Nosov (2011) examined some relevant tsunamis for which they also parametrized the sea-floor deformations.
225    It is noteworthy that those events with double amplitudes below 0.4 m were only perceptible in tide gauges, not generating a

[Figure]

**Figure 7.** Bathymetric grids used in the propagation modelling. The Grid 0, with a cell size of 500m is composed by the EMODnet 2020 bathymetry (EMODnet, 2022) and the topography by the MERIT DEM (Yamazaki et al., 2017). The Grid 1, with a cell size of 100m, is composed by the EMODnet bathymetry and the 25 m resolution topographic DEM of the IGN (CNIG, 2022). The dashed polygon labelled with CB marks the location of the Chella Bank bathymetric feature. The labels show the main localities mentioned in the text: Car, Cartagena; PM, Puerto de Mazarron; Ag, Aguilas; Crb, Carboneras; CG, Cabo de Gata; Al, Almeria; RM, Roquetas de Mar; Adr, Adra; Mo, Motril; Nj, Nerja; TM, Torre del Mar; M, Malaga; Mb, Marbella; G, Gibraltar; Ce, Ceuta; T, Tetouan; OL, Oued Laou; EJ, El Jebha; Ah, Al Hoceima; Me, Melilla; Nd, Nador; Gh, Ghazaouet; BS, Beni Saf; O, Oran.

notable impact on the coast. As can be seen in Figure 6 there is a good correspondence between the events selected to simulate with those whose double amplitudes are above 0.3 - 0.4 m.

In order to model the tsunami propagation we have resort to the highly used and validated code COMCOT (Cornell Multi-grid Coupled Tsunami) (Liu et al., 1995; Wang and Liu, 2006). This algorithm is based on the Non-linear Shallow Water Equations built over a modified leap-frog nested grids scheme.

The bathymetry used is composed of three independent sources (Figure 7). On the one hand, the bathymetric data corresponds to the EMODnet 2020 mesh (EMODnet, 2022), with a horizontal resolution of 1/16' (∼115 m). On the other hand, for the regional topography, we have used the MERIT global DEM (Yamazaki et al., 2017), with a horizontal resolution of 3" (∼90 m). For the highest resolution mesh, on the coast of Almeria, we have used the topography of the digital model of 25 m from the National Geographic Institute of Spain (CNIG, 2022). The regional mesh has been resampled with a cell size of 500 m and the local one with 100 m. **We have used open boundary conditions for the water borders, and a Manning's roughness coefficient of 0.02 when computing the inundation.**

For each of the 331 tsunami propagations we have computed the maximum elevation for a model running during 90 minutes, which is enough time for the waves to propagate through the basin and capture the wave reflections. In Figure 8 tsunami travel times are shown as well as examples of the results for three events with different magnitudes.

As expected, for the smaller magnitude events, the location of the rupture, as well as the slip distribution along the fault plane, are the determining factor in the location of the maximum wave elevations (Figure 8a-d). However, for the maximum

[Figure]

**Figure 8.** Example of maximum wave elevations obtained for three events (out of 331) with different magnitudes. The ruptures corresponding with these events are shown in Figure 2. **a and b) shows the propagation of the event 911 with $M_W$ 6.85; c and d) shows the propagation of the event 337 with $M_W$ 7.01; e and f) shows the propagation of the event 438 with $M_W$ 7.62. Dashed contours show the wave travel time in minutes.** See Figure 7 for labels of localities.

events (Figures 8e-f), in which slip occurs along the entire fault plane (see Figure 2d), the location of the maximum wave elevations are clearly determined by the morphology of the sea-floor.

245     The classical tsunami hazard deterministic approach consist on the definition of the worst-case scenario based on the dimensions of the source and the employ of a series of empirical relations to define the magnitude of the event and the average slip over the fault. Alternatively, instead of defining a single average homogeneous slip model, a set of stochastic variable slip models can be produced and analysed statistically.

[Figure]

**Figure 9. Aggregated maximum wave elevation map for a) the Alboran Sea and b) vicinity of Carboneras Fault.** See Figure 7 for labels.

[revised manuscript text omitted]

---

## Author Response (AR1)

**Comments by Julián García-Mayordomo, Geological Survey of Spain**

*Comments:*

- A proper cite of the earthquake catalogue used in figure 1 (apart from IGN-UPM (2013)) is missing. You should also cite the paper of Cabañas et al., 2015 in which the homogenization process to Mw is explained. Additionality, it is important to know that the catalogue you are using in fig 1, apart from being homogenized in size to Mw, was also declustered (IGN-UPM, 2013).

The reference and observation about declustering has been included in figure 1 caption.

- Could you extend a bit more on explaining the earthquake ruptures simulation method? I know that a proper understanding would require the reader to go to the original source, but it would be good to show an equation (for example) in which the parameters are shown (a, b,...). If it is possible.

More details on the formulation of the simulation has been added in lines 105-123.

- I assume you are using the rate and state parameters provided in Herrero-Barbero et al(2022) after a testing process that was done in that paper, is that right? If so, could you make it explicit in the manuscript; otherwise it seems that the testing process was performed in the frame of this paper.

We added a clarification in the text in lines 148-149.

- I see that references cited in the body of the manuscript are ordered alphabetically. The authors should check if that is correct according to the journal format; as references are usually ordered in increasing year of publication.

We have used the Copernicus bibliographic style for LaTeX provided by the journal. Nevertheless we will check the consistency of the references.

- I have made few more comments and suggestions for improving clarity in some parts of the text and figures right on the pdf attached. Please, in case you cannot read them (done with the adobe acrobat tool) get me back and I will pass them to you in a different format.

- I found some typos in the manuscript, also marked in the attached pdf.

We have reviewed the minor comments and suggestions from the referee and most have been included in the manuscript.

**Comments by Luis Matias, University of Lisbon**

**Major comments**

The authors apply an earthquake physical model to generated 1 Myr catalogue of earthquakes along the Carboneras Fault (CBF) and adjacent faults. The model is constrained by the assumed fault slip rates and some parameters are tuned to fit an ad-hoc Gutenberg-Richter law. The physical model generates ruptures with variable slip distribution and this feature is explored on a second part of the paper where tsunamis are generated. To our knowledge it is the first time that such physical models for earthquake generation are used in Iberia and surrounding seismically active domains. Such an effort deserves publication on itself, but additional details, and discussion must be provided to encourage the application of the model in other domains. The additional information to be provided may lead to a growth of the paper that could imply splitting it into two parts, Part 1 dedicated to the generation of the earthquake catalogue and Part 2 dedicated to the deterministic evaluation of tsunami hazard in the area. My comments will also be split according to this suggestion.

We agree with the reviewer in the fact that the first part of the paper, devoted to the physcis-based modelling, lacks of several details on the modelling and discussions on the implications and shortcomings on the asumptions made. But as is stated in the text, the presented model is already discussed and detailed in the work by Herrero-Barbero et al., 2021; which has been published in the Journal of Geophysical Research. If, as suggested by the reviewer, we publish again the details and discussions on the physics-based model, we could be incurring in self-plagiarism.

We have included additional information about the physics-based model following suggestions of both reviewers in order to be more concise on some aspects, but trying not to be excessively prolix.

Part 1:The physical model for earthquake generation

My major concern regarding this subject is the lack of relationship between observed seismicity and the Carboneras Fault. This can be inferred from Figure 1 in the paper but is made clear on figure REV-01.

As the reviewer knows, in zones of low or very low tectonic activity, the correlation between instrumental seismicity, of moderate and low magnitude, and the main faults is not direct. On the one hand, location uncertainties can be of several kilometres, and on the other, the epicenters of historical events suffer from a lack of direct observations of shallow fault ruptures. If we also take into account that the seismic cycles of these faults last thousands or tens of thousands of years, it is logical to expect that the instrumental seismicity of a few decades will not reflect the seismogenic behavior of large structures, hence the interest of physics-based models.

The paper mentions that some model parameters are tuned so that the final Gutenberg-Richter (GR) law has a b value equal to 1.0. The paper fails to give the support for this assumption and no information is provided on the a value that also characterizes the GR law. Assessing the ISC catalogue and selecting a generous area surrounding the CBF we obtain the GR law shown in figure REV-02, where the number of earthquakes is scaled to 1 Myr as in the paper.

We obtain a very high b-value, not common for convergent or transcurrent domains, showing that large magnitude events are much less frequent than found on average on the earth. This may be a feature due to the small number of events, but it deserves discussion. The thickness of the brittle layer assumed for the physical model deserves additional discussion in the light of information provided by the earthquake catalogues and deep structure studies in the area.

The calculation of a Gutenberg-Richter fit requires that the magnitudes of the events used be homogenized in order to be comparable, in addition, a completeness analysis must be done to filter the events by date and the fit should preferably be done with a maximum likelihood adjustment. Nor is it possible to extrapolate the seismicity of a few decades in a seismic cycle of thousands of years, to a behavior of hundreds of thousands or millions of years, for this reason the value of "a" of the Gutenberg Richter law, which depends on the seismic productivity is not used, but the "b" value is compared so that the distribution of the size of the events is similar to the real one.

As explained in Herrero-Barbero et al. (2021), one of the criteria for choosing the best-fit model parameters is that the b-value be close to 1, considering always the same completitude magnitude between several synthetic catalogs. This b-value is justified by the estimations in the same seismogenic zone in previous works based on instrumental seismicity (García-Mayordomo, 2005; IGN-UPM, 2013; Villamor, 2002), and is also a reference value as assumption in numerous papers of synthetic seismicity modeling (e.g., Console et al., 2017; Shaw et al., 2018). These references have been included in the text (lines 151-154).

The seismogenic crust thickness of the model is based on previous seismotectonic studies at Southeastern Spain (García-Mayordomo, 2005, Fernández-Ibañez & Soto, 2008; Mancilla et al., 2013, Grevemeyer et al., 2015). These references have been included in the text (lines 126-128).

If we compare the compute GR law with the earthquake distribution published, as shown in figure REV-03, we remark that, as suspected, the observed seismicity is much less than the modelled catalogue. This feature deserves to be discussed in the paper.

As has been answered previously, the amount of seismicity generated cannot be directly compared to the existent instrumental seismic catalogue; nevertheless details on the fit of the physics-based model, and discussions on the representativeness of the seismicity generated can be found on the Herero-Barbero et al. (2021) work. We have included this aspect in the text in lines 159-162.

Another constrain on the physical model is the average slip rate on the CBF and neighbouring faults. The authors used for the CBF the value 1.3 ± 0.2 mm/yr proposed by Echeverria et al. (2015). We quote here the Echeverria et al. (2015) sentences (CFZ = Carboneras Fault Zone): "The analysis of GPS data in the SE Betics confirmand quantify the ongoing tectonic activity of the onshore segment of the CFZ as a left-lateral strike–slip fault. For the first time, we were able to provide a quantitative measure of the present-day horizontal geodetic slip-rate of the CFZ, suggesting a maximum left-lateral motion of 1.3 ± 0.2 mm/yr. The coincidence of geologic and geodetic strike–slip rates along the CFZ, illustrates how during Quaternary its northern segment has been tectonically active and has been slipping at a rate of 1.1 to 1.5 mm/yr".

It is clearly suggested that 1.3 is a value valid only for the onshore segment of the CBF and it represents the maximum value. The use of this value deserves further discussion as well as the consequences for its variation along the fault, particularly on the ocean segment. Furthermore, our interpretation of Echeverria et al. (2015) figure 5 is that the CBF slip rate, as measured by GPS, lies between 1.0 and 2.5 mm/yr.

The only way to measure the slip-rate of faults is by means of geodetic data (GNSS or InSAR) or by means of geologic data (paleoseismology, tectonic geomorphology). From the geological point of view Moreno et al. (2015) obtained a minimum strike-slip rate of 1.31 mm/yr for a section in the onshore segment of the Carboneras fault, and a minimum dip-slip rate of 0.18 mm/yr. The offshore segment of the Carboneras fault has been extensively studied also by Moreno (2011); obtaining a minimum strike-slip rate of 1.3 mm/yr and dip-slip rates between 0.1 and 0.3 mm/yr. The geodetic data obtained to date is from the onshore segment by means of high resolution GNSS campaign

measurement and permanent stations, with the preferred value of strike-slip rate of 1.3 mm/yr (Echeverria et al., 2015). Taking into account that the onshore and offshore obtained slip rates are the same, and that the onshore geodetic data coincides with the geological data, we don't have any reason to propose any different slip-rate. We have included this short discussion on the manuscript, in lines 132-142.

All the model parameters used, as well as the associated bibliographical references, will be presented in Table 1 of the Supplementary Material. They correspond to the compilation made in Herrero-Barbero et al. (2021), shown in the Table 3 of the cited paper.

It may be relevant here for the authors to mention other sources of information on the fault slip of oceanic faults as provided by Neotectonic modelling. While Jiménez-Munt & Negredo (2003) and Cunha et al. (2012) provide slip rates estimates smaller than 0.5 mm/yr, Neres et al. (2016) show a maximum value of 1.7 mm/yr for the CBF.

Thanks to the reviewer for pointing this out. We have included the references in the text in linex 140-142.

We understand that a perfect coupling is assumed for the CBF and neighbouring faults between the kinematic constrain (slip rate) and the earthquake generation. The typical seismic coupling of major plate boundary types has been discussed by Bird and Kagan (2004). They showed that for continental convergent boundaries it lies between 0.51 and 1.00 (1.00 preferred value) and for continent transform faults it lies between 0.38 and 1.00 (0.72 preferred value). The author's choice of 1.0 deserves some discussion and the consequences of using a different value should be addressed.

It is a good observation that we have missed to mention in the text. It has been included in lines 170-171.

It seems that the generation of "characteristic earthquake" recurrence models is a feature of the physical model used. This model has not been used of PSHA and PTHA in Europe and additional discussion should be provided. Is it a model feature or is it explained by some characteristics of the CBF domain?

The pseudo-characteristic earthquake behavior arises from the model configuration. Similar results have been observed with other physics-based models (Console et al., 2021; Rafiei et al., 2022; Shaw et al., 2022). The references have been included in the text in lines 335-336.

Another feature of the physical model for earthquake recurrence applied to the CBF system is that the maximum magnitude exceeds the estimations made by several authors. It is argued that the maximum magnitude value lies at the extreme boundary of some estimates. Is it a feature of the model? Why does it happen? Some additional discussion is needed here.

The maximum magnitude exceeds that of some authors, but coincides with the estimation of others. Additional comments are included in the text in lines 182-185.

The physics-based earthquake generation model, besides the definition of the geometry (that itself deserves additional support), requires many parameters, some to be defined and others used as constrains. A list of the main parameters and information on its choice must be well presented, which is not the case on the current version of the paper. If possible, several runs of the model could be used, first to assess the random uncertainty and then to assess the epistemic uncertainty due to the choice done for some selected critical parameters. One set of the parameters, or features, of the

physics-based earthquake generation model that is not explained at all, and is relevant for the higher magnitude events, is the one that rules the multi-segment propagation.

As has been mentioned earlier, all the information required by the reviewer is already published in Herrero-Barbero et al. (2021). We consider it not necessary to repeat the same discussions and details here because we could be incurring in self-plagiarism. Additionally we have included additional information as supplementary material.

Part 2:Tsunami

Here the major comment regards the absence of any reference to previous works on tsunami hazard assessment in the area. The major study requires some reference and comparison is the NEAMTH18 (Basili et al., 2019, 2021). The results of this study can be assessed online 1 . The area investigated by the paper is shown in figure REV-04 as well as the probability results for the Almeria forecast point.

As the reviewer comments, the approximation of NEAMTH18 is probabilistic, incorporating all the potential sources estimated in the area, as such, the results are not directly comparable with our deterministic approach. Nevertheless we have mentioned NEAMTH18 in the text in lines 259-263.

**Additional comments**

On the pdf provided most of the figures are small and difficult to read.

We think that this is a problem related to the LaTeX manuscript template and in the final version the figures will be bigger.

*Lines 21-22: this ability depends on their mode of seismic rupture*

In fact, this is not the major parameter defining the potential for tsunami generation by an earthquake, as can be verified by the decision matrix adopted for the ICG/NEAMTWS (2011). The main parameters are:
- Top of the fault depth (focal depth if that information is not available)
- Location in relation to the coastline
- magnitude

Given the high directivity of tsunami propagation we may add also the strike of the structure. We suggest the authors to frame better the above-mentioned sentence.

We have to disagree with the reviewer. It is true that in the used decision matrix the rupture characteristics are not taken into account, but basically because the obtention of such rupture characteristics for an earthquake is a process that lasts several minutes in the best case, and consequently cannot be used in an emergency decision matrix. The above assertion is backed by several studies referenced in the text. It is true that the depth of the fault and the magnitude are of prime importance and we omitted it from the text as is obvious, but following the suggestion of the reviewer we have modified the text accordingly (lines 21-22).

Line 27: Although the lower capacity of strike-slip faults to generate tsunamis is a proven fact,

Mention here the strike-slip generated tsunamis in the Gloria Fault, one domain close to the one investigated in the paper and both belonging to the Nubia-Eurasia plate boundary, as presented by Baptista and Miranda (2009).

We have included the reference in the text (line 29).

Line 33: based on the simulation of tsunamis generated by ruptures of simple, rectangular

This is one occasion to mention the NEAMTHM18 model (Basili et al., 2019, 2021) that covers the investigated area and didn't use simple rectangular sources.

Ok.

In NEAMTHM18 no dynamic rupture models are used and consequently cannot be included as that in line 43.

Figure 1

The location of Almeria is missing, and it is needed.

Location of Almeria is included.

The geographic projection is "Plate Carrée" which is very unusual. Renders the comparison with other maps difficult. Why this projection was used?

The map shown in figure 1 has been generated with QGIS using a standard Mercator projection. Maybe it has been slightly modified by the vector drawing program used (Inkscape) but we think that for the purpose of the map (a location map) is precise enough.

Line 73: this fault has been proposed as source of the 1522 Almeria earthquake

This sentence is in contradiction with the location of the event in Figure 1. Clarify.

As is stated in the figure 1 caption, the location of the 1522 event is from the IGN-UPM (2013) catalog; while the proposition of the Carboneras fault as source is from Reicherter and Hubscher (2007). It has been clarified in the text as suggested (lines 79-81).

Line 79: Although the tsunami simulations done to date

Add to the list the NEAMTHM18 model (Basili et al., 2019, 2021) and comment briefly its methodology.

In NEAMTHM18 the modelling approach for faults as the Carboneras fault is done by means of "Gaussian-shape unit sources", and consequently the surface deformation is approximated based on these predefined seabottom deformations. There is no direct modelling of the Carboneras Fault itself. We have rewritten the sentence (lines 87-89).

Lines 108-109: fault depths between 8 and 12 km.

It is no clear what the authors mention as "fault depth". Is it the width of the fault? On which data is this information based?

The sentence has been partially rewritten (lines 126-128).

Figure 2

Explain in the caption the shorthand terms used, e.g., ASMF, PF, …

Ok

Line 115: Besides the input kinematic data

Our interpretation is that the authors model the ruptures on the CF and also on other faults to the NE. What is the slip rate on these faults? What was the source of information?

This information, as is stated in the text, can be found in the work describing the physics-based model of Herrero-Barbero et al. (2021). We have included the reference in the text.

Line 118: We define reference rate-and-state values based on experimental data

This experimental data was likely sampled at shallow depths. How do they apply to the expected seismogenic depths? If they are assumed to be identical on the whole fault system, what might be the consequences of this simplifying hypothesis?

The experimental data refers to fault rock mechanics laboratory tests to describe the fault zone rocks frictional behavior expected at seismogenic depths. These experimental values are the starting point before testing several synthetic catalogs with variation of the rate- and state-dependent coefficients, until achieving the best-fit values (Herrero-Barbero et al., 2021). We have clarified the sentences in the text (lines 148-151).

Line 122: rate-and-state friction parameters a=0.001 and b=0.010;

There might be a confusion with the a and b parameters of the Gutenberg-Richter law.

Yes we know, but these are the terms used in the literature. We added a caveat (lines 155-156).

Line 125: also suggested possible creeping sections in the Carboneras fault

How would this hypothesis affect the paper results? Some discussion on the assumptions and simplifications made should be provided. See main comment earlier in this document.

We added some discussion in the "Earthquake ruptures simulation" section concerning the interevent time dependence on fault coupling (lines 159-162, 170-171).

Figure 3

Given that the physics-based earthquake generation model is applied to a set of faults, it is not clear if the histogram applies only to CF or to the whole system as depicted in Figure 2. It is assumed that the a priori average slip rate of the CF is respected by the model, but a sentence presenting the a posteriori compute slip rate is needed.

As is stated in the caption the data presented is for the Carboneras fault.

Line 132: excluding the aftershocks

Since it seems that the authors are discussing the 6.5 to 7.0 magnitude interval, what is the definition of aftershocks used?

We have to apologize for the confusion, the data presented here includes aftershocks. The reference has been deleted.

Line 137: therefore the released seismic moment.

What is the shear modulus value used to compute seismic moment? Justification for that value? Is it uniform along the fault and over depth? What are the consequences of using a single average value for the modelling?

The description of the details of the model can be found on Herrero-Barbero et al. (2021). We have included here the information on the shear modulus used in line 164, which is 30 GPa, a value widely used as average shear modulus for the upper continental crust.

Line 138: The epicentres

How does the code obtain the slip initiation? Is it a feature of the code or the epicentre is computed a posteriori from the rupture distribution?

The rupture initiation in the code is governed by the rate-and-state formulation, when a "nucleation" state is reached by an element the code, spontaneously, computes the rupture propagation to the neighboring elements on a pseudo-dynamic approximation. The details can be found on the works describing the RSQsim code (Dieterich, 1995; Dieterich and Richards-Dinger, 2010; Richards-Dinger and Dieterich, 2012). Included in lines 177-180 and in an brief explanation of the physics behind the model in lines 105-123.

Lines 154-155: As the sea-floor deformation generated by the earthquake is usually transferred instantly to the elevation of the water free surface

This is not true in general, though it applies to the modelling of far source tsunamis. For locally generated tsunamis there are two effects that are not considered in the paper that deserve a comment: i) the finite compressibility acts as a filter when computing the sea surface deformation (e.g. Lotto & Dunham, 2015); ii) the horizontal movement of the sea bottom, in areas of relief, generate an initial velocity on the water that, in some circumstances, must be considered.

We have reworded the sentence (lines 198-199).

3. Tsunami modelling

There are a few general questions that must be addressed by the authors.

1) What are the boundary conditions used for the water borders?

This information has been added in line 236.

2) What are the elastic parameters used to compute the seafloor deformation? Justification?

This information has been added in line 194.

3) What happens close to the coastline? Is friction used? What are its characteristics?

This information has been added in lines 236-237.

4) Is there inundation?

This information has been added in lines 236-237.

5) How is the tsunami amplitude computed? It is recommended that the tsunami wave amplitude to be computed at cells with water depth no smaller than 50 m. The reason is explained in Kamigaichi (2011): "To represent the tsunami waveform correctly in a shallow sea area, very fine bathymetry data mesh is necessary (in a strict sense, 20 or more grid points are necessary within one wavelength [31]), and a vast time is required for the completion of such detailed calculations. To overcome this difficulty, the numerical simulation with the long-wave approximation is applied only to points which are a few to a few ten kilometers seaward from the coast ("forecast points") where sea depth is about 50m. Then, tsunami amplitude at the coast is calculated by using Green's law described in the next section."

We have modelled inundation at the coast and consequently there is no use of the green's law.

Line 204: maximum wave elevation

The meaning of this parameter must be well explained. See my previous comment.

I think I don't fully understand the reviewer concern with the term. It is the widely used term to describe the maximum elevation reached by the free water surface at a point of the calculation grid on a propagation.

Line 211: relevant local inundations

It is not explained how "maximum elevation" is converted to inundation. See previous comments.

The inundation is computed by means of the COMCOT numerical model.

Figure 8

Explain the dashed contours.

Added to the caption.

Line 218: have been taken for the 5 m depth isobath

This explanation should have been provided earlier. Given previous comments 5 m seems not appropriate. What happens if there is no cell at 5 m depth? Use the Green law to convert it to 5 m?

As the inundation is computed there is no need to use the Green law.

Lines 224 – 234

Given that a single, randomly generated, catalogue was used, I fail to see the relevance of the discussion on these details of the tsunami amplitude histogram.

Would another catalogue generate the same features?

The range in maximum magnitudes would be probably the same as it depends on the dimensions of the structures. If we vary the slip rate or the seismic coupling, the interevent times would be different, and consequently the statistical distribution of maximum elevations and inundations.

The crucial point here is that, when dealing with deterministic approaches usually a single worst case scenario is used. In the best case a set of worst case scenarios considering the uncertainties is done. With these models we can obtain a physically coherent seismic behavior through hundreds of seismic cycles that produce hundreds or thousands of destructive events that can be modeled to characterize not only the worst-case scenario, but also characterize scenarios linked to probabilities of exceedance or return periods.

We have extended our reasoning in the text in lines 304-312.

Line 239: generate locally damaging tsunamis

Define "local" tsunamis. The term "local" has a very specific meaning in the tsunami warning systems (ICG/NEAMTWS ,2011)

We think that the term "locally damaging tsunamis" accompanied with the maps showing the areas where the maximum waves are expected is clear enough. The terminology used in warning systems are crucial for the development of warning systems, but not for the object of our study.

Line 243: If we compare the results of this work with previous results

Given that the "frequency" of tsunamis was mentioned in line 241, comments on the NEAMTHM18 model (Basili et al., 2019, 2021) are appropriate here.

We refer here to the results of our work as we think is clearly stated in the text.

Line 248: Mw 7.62, which is close to the maximum magnitude proposed by Moreno (2011).

This value and reference were not mentioned in the introduction and they are relevant.

It has been included in line 84.

Lines 252 – 253: On the other hand, the rake used in our models is 10° while Alvarez-Gomez et al. (2011a) used 15o and Gomez de la Peña et al. (2022) used 0°.

What are the consequences of the uncertainty on the rake to the paper results?

Additional comments have been included in lines 302-303.

Lines 264 – 265: we can obtain the maximum magnitudes in a robust manner from a statistical point of view.

Given the larger number of simplifications and approximations used in the physics-based earthquake generation model, given that a single catalogue was generated without assessing aleatoric and epistemic uncertainties, I cannot classify the results as "robust", though deserving to be published.

**We have reworded the phrase.**

Lines 270 – 271: relationship between the size of the earthquake rupture and the slip;

Not shown in the paper. Show as supplement?

**The earthquake ruptures are included now as supplementary data.**

Line 278: would reflect a GR relation

In fact, one of the most frequent earthquake recurrence laws is the truncated GR relation, not the simple (and open) GR law.

**We are referring to both flavors when mentioning the Gutenberg-Richter relation.**

Line 285: see supplementary models

These are not available on the documentation provided.

**Our mistake. Now is included.**

Line 295: the simplified model overestimates

It should be mentioned that nowadays the common procedure is to taper the uniform slip at the borders of the rectangular faults (e.g. Davies and Griffin, 2018). Given this, the comparison between tsunamis generated by irregular and uniform slip faults is unfair, for the tips of the fault as mentioned in the text.

**Additional comments are included in lines 359-360.**

Figure 13

The labels mentioned in the caption cannot be seen on the figure. Too small?

**We think that this is a problem related to the LaTeX manuscript template and in the final version the figures will be bigger.**

Line 309: allows a more robust characterization of the scenarios

Given the larger number of simplifications and approximations used in the physics-based earthquake generation model, given that a single catalogue was generatedwithout assessing aleatoric and epistemic uncertainties, I cannot classify the results as "robust", though deserving to be published.

**We have reworded the phrase.**

Figure 14

The labels mentioned in the caption cannot be seen on the figure. Too small?

We think that this is a problem related to the LaTeX manuscript template and in the final version the figures will be bigger.

Lines 316, 317: which allows a robust implementation of uncertainty estimation
Given the larger number of simplifications and approximations used in the physics-based earthquake generation model, given that a single catalogue was generated without assessing aleatoric and epistemic uncertainties, I cannot classify the results as "robust", though deserving to be published.

We have reworded the phrase.

Lines 332, 333: The implementation of these methodologies in the Probabilistic Tsunami Hazard Analyses (PTHA) is a logical and necessary step.

Comment/discussion of the NEAMTHM18 model (Basili et al., 2019, 2021) are needed here since they apply to the same area discussed in the paper.

We do not see the necessity of including in our conclusions this reference again, as has been now included several times throughout the text.

Line 337: GMT (Wessel et aI., 2013) has been used to

Given this information we do not understand the use of the "Plate Carrée" projection in Figure 1.

As is stated in the text: "GMT (Wessel et al., 2013) has been used to perform some calculations and to produce most of the figures." not all. The map shown in figure 1 has been generated with QGIS using a standard Mercator projection. Maybe it has been slightly modified by the vector drawing program used (Inkscape) but we think that for the purpose of the map (a location map) is precise enough.

---

## Editor Decision (ED1)

**NHESS-2022-186 Seismogenic potential and tsunami threat of the strike-slip Carboneras Fault in the Western Mediterranean from physics-based earthquake simulations**

José A. Álvarez-Gómez, Paula Herrero-Barbero, and José J. Martínez-Díaz

**Comments on revised version by Luis Matias, University of Lisbon**

**Recommendation**

It is my recommendation that the work is ready for publication with minor corrections that are detailed below. The major concern on the current version has to do with the tsunami modelling procedure. It was not clear for me on the first version that the modelling included inundation. This misunderstanding raised several questions that were not appropriate if inundation was computed. Inundation requires a proper DTM for the land mass and this source of information is still missing on the current version. As a broad comment I suggest the authors to clarify this issue.

**Major comments**

In the following I make additional comments (in blue) on top of the *original comments (in black and italic)* and the author's reply (in black). Only those comments that merit additional changes to the manuscript are mentioned.

*Part 1:The physical model for earthquake generation*

*My major concern regarding this subject is the lack of relationship between observed seismicity and the Carboneras Fault. This can be inferred from Figure 1 in the paper but is made clear on figure REV-01 (not repeated here).*

As the reviewer knows, in zones of low or very low tectonic activity, the correlation between

instrumental seismicity, of moderate and low magnitude, and the main faults is not direct. On the one hand, location uncertainties can be of several kilometres, and on the other, the epicenters of historical events suffer from a lack of direct observations of shallow fault ruptures. If we also take into account that the seismic cycles of these faults last thousands or tens of thousands of years, it is logical to expect that the instrumental seismicity of a few decades will not reflect the seismogenic behavior of large structures, hence the interest of physics-based models.

*I agree with the author's comments, but I am in favour that this information should somehow appear in the manuscript to emphasize the use of the methodology to other slow deforming regions.*

*The paper mentions that some model parameters are tuned so that the final Gutenberg-Richter (GR) law has a b value equal to 1.0. The paper fails to give the support for this assumption and no information is provided on the a value that also characterizes the GR law. Assessing the ISC catalogue and selecting a generous area surrounding the CBF we obtain the GR law shown in figure REV-02, where the number of earthquakes is scaled to 1 Myr as in the paper.*

*We obtain a very high b-value, not common for convergent or transcurrent domains, showing that large magnitude events are much less frequent than found on average on the earth. This may be a feature due to the small number of events, but it deserves discussion. The thickness of the brittle layer assumed for the physical model deserves additional discussion in the light of information provided by the earthquake catalogues and deep structure studies in the area.*

The calculation of a Gutenberg-Richter fit requires that the magnitudes of the events used be homogenized in order to be comparable, in addition, a completeness analysis must be done to filter the events by date and the fit should preferably be done with a maximum likelihood adjustment. Nor is it possible to extrapolate the seismicity of a few decades in a seismic cycle of thousands of years, to a behavior of hundreds of thousands or millions of years, for this reason the value of "a" of the Gutenberg Richter law, which depends on the seismic productivity is not used, but the "b" value is compared so that the distribution of the size of the events is similar to the real one. As explained in Herrero-Barbero et al. (2021), one of the criteria for choosing the best-fit model parameters is that the b-value be close to 1, considering always the same completitude magnitude between several synthetic catalogs. This b-value is justified by the estimations in the same seismogenic zone in previous works based on instrumental seismicity (García-Mayordomo, 2005; IGN-UPM, 2013; Villamor, 2002), and is also a reference value as assumption in numerous papers of synthetic seismicity modeling (e.g., Console et al., 2017; Shaw et al., 2018). These references have been included in the text (lines 151-154). The seismogenic crust thickness of the model is based on previous seismotectonic studies at Southeastern Spain (García-Mayordomo, 2005, Fernández-Ibañez & Soto, 2008; Mancilla et al., 2013, Grevemeyer et al., 2015). These references have been included in the text (lines 126-128).

In my opinion, some short version of this information should appear on the manuscript, not for the benefit of the reviewer, but for the benefit of the reader.

**Additional comments**

*The geographic projection is "Plate Carrée" which is very unusual. Renders the comparison with other maps difficult. Why this projection was used?*

The map shown in figure 1 has been generated with QGIS using a standard Mercator projection. Maybe it has been slightly modified by the vector drawing program used (Inkscape) but we think that for the purpose of the map (a location map) is precise enough.

The figure on the next page shows the geographical area of figure 1 as presented in the manuscript and as plot with Mercator projection by GMT. I added a rectangle 1º by 1º on both plots. This shows that the projection used in the manuscript is not Mercator as claimed. I do not suggest redoing the figure, just mention on the caption the geographical projection used, for the benefit of the reader.

[Figure]

*Lines 154-155: As the sea-floor deformation generated by the earthquake is usually transferred instantly to the 155 elevation of the water free surface*

*This is not true in general, though it applies to the modelling of far source tsunamis. For locally generated tsunamis there are two effects that are not considered in the paper that deserve a comment: i) the finite compressibility acts as a filter when computing the sea surface deformation (e.g. Lotto & Dunham, 2015); ii) the horizontal movement of the sea bottom, in areas of relief, generate an initial velocity on the water that, in some circumstances, must be considered.*

We have reworded the sentence (lines 198-199).

I see that the authors address (ii) above but not (i). It is a detail that is missed in many tsunami simulations but for this manuscript its relevance may be considered second order.

*3. Tsunami modelling*

*5) How is the tsunami amplitude computed? It is recommended that the tsunami wave amplitude to be computed at cells with water depth no smaller than 50 m. The reason is explained in Kamigaichi (2011): "To represent the tsunami waveform correctly in a shallow sea area, very fine bathymetry data mesh is necessary (in a strict sense, 20 or more grid points are necessary within one wave-length [31]), and a vast time is required for the completion of such detailed calculations. To overcome this difficulty, the numerical simulation with the long-wave approximation is applied only to points which are a few to a few ten kilometers seaward from the coast ("forecast points") where sea depth is about 50m. Then, tsunami amplitude at the coast is calculated by using Green's law described in the next section."*

We have modelled inundation at the coast and consequently there is no use of the green's law.

This is not completely satisfying. The figures presented in the manuscript show only "maximum elevation" but no inundation, generating the question in my original comment. Is it possible to show one inundation map as supplementary material?

*Line 204: maximum wave elevation*

*The meaning of this parameter must be well explained. See my previous comment.*

I think I don't fully understand the reviewer concern with the term. It is the widely used term to describe the maximum elevation reached by the free water surface at a point of the calculation grid on a propagation.

This comment is indeed true if the tsunami propagation extends to the inundation phase. The mentions of "inundation" in the manuscript are scarce:

*Lines 232-233: … a Manning's roughness coefficient of 0.02 when computing the inundation.*

*Lines 250-251: and with relevant inundations, in the Almerian coast (Figure 9).*

Figure 9 only shows maximum elevation, not inundation that we might see. At least it is not mentioned in the caption.

*Line 253: and with relevant inundations, in the Almerian coast (Figure 9).*

*Line 303: and consequently the statistical distribution of maximum elevations and inundations.*

I believe that the computation of inundation in tsunami modelling should be clarified or emphasized, given its relevance for the discussion. For inundation the authors need a detailed DTM for the land mass but no mention to it is found on the manuscript.

*Line 211: relevant local inundations*

*It is not explained how "maximum elevation" is converted to inundation. See previous comments.*

The inundation is computed by means of the COMCOT numerical model.

The only reference to COMCOT is found on line 224: In order to model the tsunami propagation we have resort to the highly used and validated code COMCOT.

In my opinion "tsunami propagation" is not equivalent to the computation of "tsunami inundation" which is more demanding computationally and requires new detailed datasets not mentioned in the manuscript.

---

## Author Response (AR2)

**Comments by Luis Matias, University of Lisbon**

Major comments

…
I agree with the author's comments, but I am in favour that this information should somehow appear in the manuscript to emphasize the use of the methodology to other slow deforming regions.

Following the suggestion we have included this in lines 80 – 83.

…
In my opinion, some short version of this information should appear on the manuscript, not for the benefit of the reviewer, but for the benefit of the reader.

The information provided in the first answer to the reviewer is included in lines from 141 to 154.

…
The figure on the next page shows the geographical area of figure 1 as presented in the manuscript and as plot with Mercator projection by GMT. I added a rectangle 1o by 1o on both plots. This shows that the projection used in the manuscript is not Mercator as claimed. I do not suggest redoing the figure, just mention on the caption the geographical projection used, for the benefit of the reader.

We added a note on Figure 1 caption.

…
I see that the authors address (ii) above but not (i). It is a detail that is missed in many tsunami simulations but for this manuscript its relevance may be considered second order.

We have included a note on this issue on lines 229-231, and included the reference to Lotto and Dunham (2015).

…
This is not completely satisfying. The figures presented in the manuscript show only "maximum elevation" but no inundation, generating the question in my original comment. Is it possible to show one inundation map as supplementary material?

We have included new subfigures in figure 8 showing a more detailed map of some results including inundation.

…
This comment is indeed true if the tsunami propagation extends to the inundation phase. The mentions of "inundation" in the manuscript are scarce:
Lines 232-233: ... a Manning's roughness coefficient of 0.02 when computing the inundation.
Lines 250-251: and with relevant inundations, in the Almerian coast (Figure 9).
Figure 9 only shows maximum elevation, not inundation that we might see. At least it is not mentioned in the caption.
Line 253: and with relevant inundations, in the Almerian coast (Figure 9).
Line 303: and consequently the statistical distribution of maximum elevations and inundations.

I believe that the computation of inundation in tsunami modelling should be clarified or emphasized, given its relevance for the discussion. For inundation the authors need a detailed DTM for the land mass but no mention to it is found on the manuscript.

We have included additional information on tsunami inundation on lines 227, 235. The original resolution for the topography is a DTM of 25m from the National Geographic institute of Spain as is mentioned and referenced in line 236.
…
The inundation is computed by means of the COMCOT numerical model.
The only reference to COMCOT is found on line 224: In order to model the tsunami propagation we have resort to the highly used and validated code COMCOT.
In my opinion "tsunami propagation" is not equivalent to the computation of "tsunami inundation" which is more demanding computationally and requires new detailed datasets not mentioned in the manuscript.

As has been said before the details on the inundation DTM is included in the text. Also new mentions to the inundation have been included. In lines 227, 235 and figure 8 caption.